# Efficiently Modeling Long Sequences with Structured State Spaces

**Albert Gu & Karan Goel & Christopher Ré**
Department of Computer Science, Stanford University
{albertgu,krng}@stanford.edu, chrismre@cs.stanford.edu

## Abstract

A central goal of sequence modeling is designing a single principled model that can address sequence data across a range of modalities and tasks, particularly on long-range dependencies. Although conventional models including RNNs, CNNs, and Transformers have specialized variants for capturing long dependencies, they still struggle to scale to very long sequences of 10000 or more steps. A promising recent approach proposed modeling sequences by simulating the fundamental state space model (SSM) $x'(t) = Ax(t) + Bu(t), y(t) = Cx(t) + Du(t)$, and showed that for appropriate choices of the state matrix $A$, this system could handle long-range dependencies mathematically and empirically. However, this method has prohibitive computation and memory requirements, rendering it infeasible as a general sequence modeling solution. We propose the Structured State Space (S4) sequence model based on a new parameterization for the SSM, and show that it can be computed much more efficiently than prior approaches while preserving their theoretical strengths. Our technique involves conditioning $A$ with a low-rank correction, allowing it to be diagonalized stably and reducing the SSM to the well-studied computation of a Cauchy kernel. S4 achieves strong empirical results across a diverse range of established benchmarks, including (i) 91% accuracy on sequential CIFAR-10 with no data augmentation or auxiliary losses, on par with a larger 2-D ResNet, (ii) substantially closing the gap to Transformers on image and language modeling tasks, while performing generation $60\times$ faster (iii) SoTA on every task from the Long Range Arena benchmark, including solving the challenging Path-X task of length 16k that all prior work fails on, while being as efficient as all competitors.[1]

## 1 Introduction

A central problem in sequence modeling is efficiently handling data that contains long-range dependencies (LRDs). Real-world time-series data often requires reasoning over tens of thousands of time steps, while few sequence models address even thousands of time steps. For instance, results from the long-range arena (LRA) benchmark (Tay et al., 2021) highlight that sequence models today perform poorly on LRD tasks, including one (Path-X) where no model performs better than random guessing.

Since LRDs are perhaps the foremost challenge for sequence models, all standard model families such as continuous-time models (CTMs), RNNs, CNNs, and Transformers include many specialized variants designed to address them. Modern examples include orthogonal and Lipschitz RNNs (Arjovsky et al., 2016; Erichson et al., 2021) to combat vanishing gradients, dilated convolutions to increase context size (Bai et al., 2018; Oord et al., 2016), and an increasingly vast family of efficient Transformers that reduce the quadratic dependence on sequence length (Katharopoulos et al., 2020; Choromanski et al., 2020). Despite being designed for LRDs, these solutions still perform poorly on challenging benchmarks such as LRA (Tay et al., 2021) or raw audio classification (Gu et al., 2021).

An alternative approach to LRDs was recently introduced based on the **state space model (SSM)** (Fig. 1). SSMs are a foundational scientific model used in fields such as control theory, computational neuroscience, and many more, but have not been applicable to deep learning for concrete theoretical reasons. In particular, Gu et al. (2021) showed that deep SSMs actually struggle even on simple tasks, but can perform exceptionally well when equipped with special state matrices $\boldsymbol{A}$ recently derived

---

[1]Code is publicly available at https://github.com/HazyResearch/state-spaces.

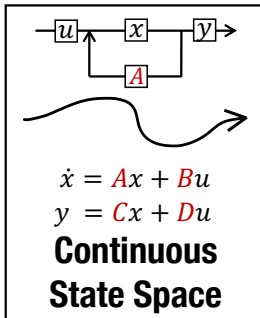
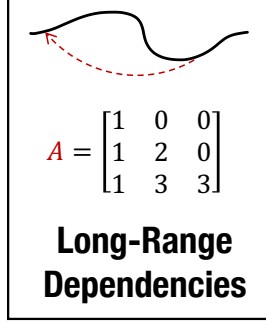
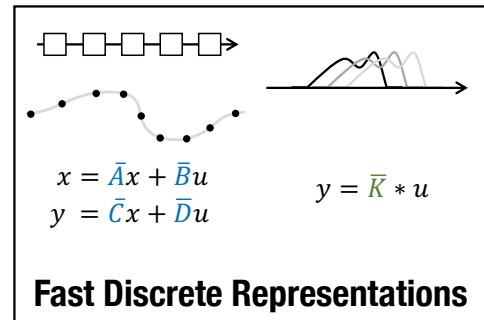

Figure 1: (**Left**) State Space Models (SSM) parameterized by matrices $\boldsymbol{A}, \boldsymbol{B}, \boldsymbol{C}, \boldsymbol{D}$ map an input signal $u(t)$ to output $y(t)$ through a latent state $x(t)$. (**Center**) Recent theory on continuous-time memorization derives special $\boldsymbol{A}$ matrices that allow SSMs to capture LRDs mathematically and empirically. (**Right**) SSMs can be computed either as a recurrence (left) or convolution (right). However, materializing these conceptual views requires utilizing different representations of its parameters (red, blue, green) which are very expensive to compute. S4 introduces a novel parameterization that efficiently swaps between these representations, allowing it to handle a wide range of tasks, be efficient at both training and inference, and excel at long sequences.

to solve a problem of continuous-time memorization (Voelker et al., 2019; Gu et al., 2020a). Their Linear State Space Layer (LSSL) conceptually unifies the strengths of CTM, RNN and CNN models, and provides a proof of concept that deep SSMs can address LRDs in principle.

Unfortunately, the LSSL is infeasible to use in practice because of prohibitive computation and memory requirements induced by the state representation. For state dimension $N$ and sequence length $L$, computing the latent state requires $O(N^2 L)$ operations and $O(NL)$ space – compared to a $\Omega(L + N)$ lower bound for both. Thus for reasonably sized models (e.g. $N = 256$ in Gu et al. (2021)), the LSSL uses orders of magnitude more memory than comparably-sized RNNs or CNNs. Although theoretically efficient algorithms for the LSSL were proposed, we show that these are numerically unstable. In particular, the special $\boldsymbol{A}$ matrix is highly non-normal in the linear algebraic sense, which prevents the application of conventional algorithmic techniques. Consequently, although the LSSL showed that SSMs have strong performance, they are currently computationally impractical as a general sequence modeling solution.

In this work, we introduce the **Structured State Space (S4)** sequence model based on the SSM that solves the critical computational bottleneck in previous work. Technically, S4 reparameterizes the structured state matrices $\boldsymbol{A}$ appearing in Voelker et al. (2019); Gu et al. (2020a) by decomposing them as the sum of a low-rank and skew-symmetric term. Additionally, instead of expanding the standard SSM in coefficient space, we compute its truncated generating function in frequency space, which can be simplified into a multipole-like evaluation. Combining these two ideas, we show that the low-rank term can be corrected by the Woodbury identity while the skew-symmetric term can be diagonalized stably, ultimately reducing to a well-studied and theoretically stable Cauchy kernel (Pan, 2001; 2017). This results in $\tilde{O}(N + L)$ computation and $O(N + L)$ memory usage, which is essentially tight for sequence models. Compared to the LSSL, S4 is up to $30\times$ faster with $400\times$ less memory usage, while exceeding the LSSL's performance empirically.

Empirically, S4 significantly advances the state-of-the-art for LRD. On the LRA benchmark for efficient sequence models, S4 is as fast as all baselines while outperforming them by $20+$ points on average. S4 is the first model to solve the difficult LRA Path-X task (length-16384), achieving **88% accuracy compared to 50% random guessing** for all prior work. On speech classification with length-16000 sequences, S4 halves the test error ($1.7\%$) of specialized Speech CNNs – by contrast, all RNN and Transformer baselines fail to learn ($\geq 70\%$ error).

**Towards a general-purpose sequence model.**  Beyond LRD, a broad goal of machine learning is to develop a single model that can be used across a wide range of problems. Models today are typically specialized to solve problems from a particular domain (e.g. images, audio, text, time-series), and enable a narrow range of capabilities (e.g. efficient training, fast generation, handling irregularly sampled data). This specialization is typically expressed via domain-specific preprocessing, inductive biases, and architectures. Sequence models provide a general framework for solving many of these problems with reduced specialization – e.g. Vision Transformers for image classification with less

2D information (Dosovitskiy et al., 2020). However, most models such as Transformers generally still require substantial specialization per task to achieve high performance.

Deep SSMs in particular have conceptual strengths that suggest they may be promising as a general sequence modeling solution. These strengths include a principled approach to handling LRDs, as well as the ability to move between continuous-time, convolutional, and recurrent model representations, each with distinct capabilities (Fig. 1). Our technical contributions enable SSMs to be applied successfully to a varied set of benchmarks with minimal modification:

- *Large-scale generative modeling.* On CIFAR-10 density estimation, S4 is competitive with the best autoregressive models (2.85 bits per dim). On WikiText-103 language modeling, S4 substantially closes the gap to Transformers (within $0.8$ perplexity), setting SoTA for attention-free models.

- *Fast autoregressive generation.* Like RNNs, S4 can use its latent state to perform $60\times$ faster pixel/token generation than standard autoregressive models on CIFAR-10 and WikiText-103.

- *Sampling resolution change.* Like specialized CTMs, S4 can adapt to changes in time-series sampling frequency without retraining, e.g. at $0.5\times$ frequency on speech classification.

- *Learning with weaker inductive biases.* With no architectural changes, S4 surpasses Speech CNNs on speech classification, outperforms the specialized Informer model on time-series forecasting problems, and matches a 2-D ResNet on sequential CIFAR with over $90\%$ accuracy.

## 2 BACKGROUND: STATE SPACES

Sections 2.1 to 2.4 describe the four properties of SSMs in Fig. 1: the classic continuous-time representation, addressing LRDs with the HiPPO framework, the discrete-time recurrent representation, and the parallelizable convolution representation. In particular, Section 2.4 introduces the SSM convolution kernel $\overline{\boldsymbol{K}}$, which is the focus of our theoretical contributions in Section 3.

### 2.1 STATE SPACE MODELS: A CONTINUOUS-TIME LATENT STATE MODEL

The state space model is defined by the simple equation (1). It maps a 1-D input signal $u(t)$ to an $N$-D latent state $x(t)$ before projecting to a 1-D output signal $y(t)$.

$$
\begin{aligned}
x'(t) &= \boldsymbol{A}x(t) + \boldsymbol{B}u(t) \\
y(t) &= \boldsymbol{C}x(t) + \boldsymbol{D}u(t)
\end{aligned}
\tag{1}
$$

SSMs are broadly used in many scientific disciplines and related to latent state models such as Hidden Markov Models (HMM). Our goal is to simply use the SSM as a black-box representation in a deep sequence model, where $\boldsymbol{A}, \boldsymbol{B}, \boldsymbol{C}, \boldsymbol{D}$ are parameters learned by gradient descent. For the remainder of this paper, we will omit the parameter $\boldsymbol{D}$ for exposition (or equivalently, assume $\boldsymbol{D} = 0$) because the term $\boldsymbol{D}u$ can be viewed as a skip connection and is easy to compute.

### 2.2 ADDRESSING LONG-RANGE DEPENDENCIES WITH HiPPO

Prior work found that the basic SSM (1) actually performs very poorly in practice. Intuitively, one explanation is that linear first-order ODEs solve to an exponential function, and thus may suffer from gradients scaling exponentially in the sequence length (i.e., the vanishing/exploding gradients problem (Pascanu et al., 2013)). To address this problem, the LSSL leveraged the HiPPO theory of continuous-time memorization (Gu et al., 2020a). HiPPO specifies a class of certain matrices $\boldsymbol{A} \in \mathbb{R}^{N \times N}$ that when incorporated into (1), allows the state $x(t)$ to memorize the history of the input $u(t)$. The most important matrix in this class is defined by equation (2), which we will call the HiPPO matrix. For example, the LSSL found that simply modifying an SSM from a random matrix $\boldsymbol{A}$ to equation (2) improved its performance on the sequential MNIST benchmark from $60\%$ to $98\%$.

$$
(\textbf{HiPPO Matrix}) \qquad \boldsymbol{A}_{nk} = -
\begin{cases}
(2n+1)^{1/2}(2k+1)^{1/2} & \text{if } n > k \\
n+1 & \text{if } n = k \\
0 & \text{if } n < k
\end{cases}
\tag{2}
$$

## 2.3 Discrete-time SSM: The Recurrent Representation

To be applied on a discrete input sequence $(u_0, u_1, \dots)$ instead of continuous function $u(t)$, (1) must be discretized by a **step size** $\Delta$ that represents the resolution of the input. Conceptually, the inputs $u_k$ can be viewed as sampling an implicit underlying continuous signal $u(t)$, where $u_k = u(k\Delta)$.

To discretize the continuous-time SSM, we follow prior work in using the bilinear method (Tustin, 1947), which converts the state matrix $\boldsymbol{A}$ into an approximation $\overline{\boldsymbol{A}}$ . The discrete SSM is

$$
\begin{aligned}
x_k &= \overline{\boldsymbol{A}} x_{k-1} + \overline{\boldsymbol{B}} u_k & \overline{\boldsymbol{A}} &= (\boldsymbol{I} - \Delta/2 \cdot \boldsymbol{A})^{-1}(\boldsymbol{I} + \Delta/2 \cdot \boldsymbol{A}) \\
y_k &= \overline{\boldsymbol{C}} x_k & \overline{\boldsymbol{B}} &= (\boldsymbol{I} - \Delta/2 \cdot \boldsymbol{A})^{-1} \Delta \boldsymbol{B} & \overline{\boldsymbol{C}} &= \boldsymbol{C}.
\end{aligned}
\tag{3}
$$

Equation (3) is now a *sequence-to-sequence* map $u_k \mapsto y_k$ instead of function-to-function. Moreover the state equation is now a recurrence in $x_k$, allowing the discrete SSM to be computed like an RNN. Concretely, $x_k \in \mathbb{R}^N$ can be viewed as a *hidden state* with transition matrix $\overline{\boldsymbol{A}}$.

Notationally, throughout this paper we use $\overline{\boldsymbol{A}}, \overline{\boldsymbol{B}}, \dots$ to denote discretized SSM matrices defined by (3). Note that these matrices are a function of both $\boldsymbol{A}$ as well as a step size $\Delta$; we suppress this dependence for notational convenience when it is clear.

## 2.4 Training SSMs: The Convolutional Representation

The recurrent SSM (3) is not practical for training on modern hardware due to its sequentiality. Instead, there is a well-known connection between linear time-invariant (LTI) SSMs such as (1) and continuous convolutions. Correspondingly, (3) can actually be written as a discrete convolution.

For simplicity let the initial state be $x_{-1} = 0$. Then unrolling (3) explicitly yields

$$
\begin{aligned}
x_0 &= \overline{\boldsymbol{B}} u_0 & x_1 &= \overline{\boldsymbol{A}\boldsymbol{B}} u_0 + \overline{\boldsymbol{B}} u_1 & x_2 &= \overline{\boldsymbol{A}}^2 \overline{\boldsymbol{B}} u_0 + \overline{\boldsymbol{A}\boldsymbol{B}} u_1 + \overline{\boldsymbol{B}} u_2 & \dots \\
y_0 &= \overline{\boldsymbol{C}\boldsymbol{B}} u_0 & y_1 &= \overline{\boldsymbol{C}\boldsymbol{A}\boldsymbol{B}} u_0 + \overline{\boldsymbol{C}\boldsymbol{B}} u_1 & y_2 &= \overline{\boldsymbol{C}\boldsymbol{A}}^2 \overline{\boldsymbol{B}} u_0 + \overline{\boldsymbol{C}\boldsymbol{A}\boldsymbol{B}} u_1 + \overline{\boldsymbol{C}\boldsymbol{B}} u_2 & \dots
\end{aligned}
$$

This can be vectorized into a convolution (4) with an explicit formula for the convolution kernel (5).

$$
\begin{aligned}
y_k &= \overline{\boldsymbol{C}\boldsymbol{A}}^k \overline{\boldsymbol{B}} u_0 + \overline{\boldsymbol{C}\boldsymbol{A}}^{k-1} \overline{\boldsymbol{B}} u_1 + \dots + \overline{\boldsymbol{C}\boldsymbol{A}\boldsymbol{B}} u_{k-1} + \overline{\boldsymbol{C}\boldsymbol{B}} u_k \\
y &= \overline{\boldsymbol{K}} * u.
\end{aligned}
\tag{4}
$$

$$
\overline{\boldsymbol{K}} \in \mathbb{R}^L := \mathcal{K}_L(\overline{\boldsymbol{A}}, \overline{\boldsymbol{B}}, \overline{\boldsymbol{C}}) := \left( \overline{\boldsymbol{C}\boldsymbol{A}}^i \overline{\boldsymbol{B}} \right)_{i \in [L]} = (\overline{\boldsymbol{C}\boldsymbol{B}}, \overline{\boldsymbol{C}\boldsymbol{A}\boldsymbol{B}}, \dots, \overline{\boldsymbol{C}\boldsymbol{A}}^{L-1} \overline{\boldsymbol{B}}).
\tag{5}
$$

In other words, equation (4) is a single (non-circular) convolution and can be computed very efficiently with FFTs, *provided* that $\overline{\boldsymbol{K}}$ is known. However, computing $\overline{\boldsymbol{K}}$ in (5) is non-trivial and is the focus of our technical contributions in Section 3. We call $\overline{\boldsymbol{K}}$ the **SSM convolution kernel** or filter.

## 3 Method: Structured State Spaces (S4)

Our technical results focus on developing the S4 parameterization and showing how to efficiently compute all views of the SSM (Section 2): the continuous representation $(\boldsymbol{A}, \boldsymbol{B}, \boldsymbol{C})$ (1), the recurrent representation $(\overline{\boldsymbol{A}}, \overline{\boldsymbol{B}}, \overline{\boldsymbol{C}})$ (3), and the convolutional representation $\overline{\boldsymbol{K}}$ (4).

Section 3.1 motivates our approach, which is based on the linear algebraic concepts of conjugation and diagonalization, and discusses why the naive application of this approach does not work. Section 3.2 gives an overview of the key technical components of our approach and formally defines the S4 parameterization. Section 3.3 sketches the main results, showing that S4 is asymptotically efficient (up to log factors) for sequence models. Proofs are in Appendices B and C.

## 3.1 Motivation: Diagonalization

The fundamental bottleneck in computing the discrete-time SSM (3) is that it involves repeated matrix multiplication by $\overline{\boldsymbol{A}}$. For example, computing (5) naively as in the LSSL involves $L$ successive multiplications by $\overline{\boldsymbol{A}}$, requiring $O(N^2 L)$ operations and $O(NL)$ space.

To overcome this bottleneck, we use a structural result that allows us to simplify SSMs.

---

**Algorithm 1** S4 CONVOLUTION KERNEL (SKETCH)

---

**Input:** S4 parameters $\mathbf{\Lambda}, \boldsymbol{P}, \boldsymbol{Q}, \boldsymbol{B}, \boldsymbol{C} \in \mathbb{C}^N$ and step size $\Delta$
**Output:** SSM convolution kernel $\overline{\boldsymbol{K}} = \mathcal{K}_L(\overline{\boldsymbol{A}}, \overline{\boldsymbol{B}}, \overline{\boldsymbol{C}})$ for $\boldsymbol{A} = \mathbf{\Lambda} - \boldsymbol{P}\boldsymbol{Q}^*$ (equation (5))

1: $\widetilde{\boldsymbol{C}} \leftarrow \left(\boldsymbol{I} - \overline{\boldsymbol{A}}^L\right)^* \overline{\boldsymbol{C}}$           ▷ Truncate SSM generating function (SSMGF) to length $L$

2: $\begin{bmatrix} k_{00}(\omega) & k_{01}(\omega) \\ k_{10}(\omega) & k_{11}(\omega) \end{bmatrix} \leftarrow \left[\widetilde{\boldsymbol{C}}\ \boldsymbol{Q}\right]^* \left(\frac{2}{\Delta}\frac{1-\omega}{1+\omega} - \mathbf{\Lambda}\right)^{-1} [\boldsymbol{B}\ \boldsymbol{P}]$      ▷ Black-box Cauchy kernel

3: $\hat{\boldsymbol{K}}(\omega) \leftarrow \frac{2}{1+\omega}\left[k_{00}(\omega) - k_{01}(\omega)(1 + k_{11}(\omega))^{-1}k_{10}(\omega)\right]$      ▷ Woodbury Identity

4: $\hat{\boldsymbol{K}} = \{\hat{\boldsymbol{K}}(\omega) : \omega = \exp(2\pi i \frac{k}{L})\}$      ▷ Evaluate SSMGF at all roots of unity $\omega \in \Omega_L$

5: $\overline{\boldsymbol{K}} \leftarrow \text{iFFT}(\hat{\boldsymbol{K}})$      ▷ Inverse Fourier Transform

---

**Lemma 3.1.** *Conjugation is an equivalence relation on SSMs* $(\boldsymbol{A}, \boldsymbol{B}, \boldsymbol{C}) \sim (\boldsymbol{V}^{-1}\boldsymbol{A}\boldsymbol{V}, \boldsymbol{V}^{-1}\boldsymbol{B}, \boldsymbol{C}\boldsymbol{V})$.

*Proof.* Write out the two SSMs with state denoted by $x$ and $\tilde{x}$ respectively:

$$x' = \boldsymbol{A}x + \boldsymbol{B}u \qquad\qquad \tilde{x}' = \boldsymbol{V}^{-1}\boldsymbol{A}\boldsymbol{V}\tilde{x} + \boldsymbol{V}^{-1}\boldsymbol{B}u$$
$$y = \boldsymbol{C}x \qquad\qquad\qquad y = \boldsymbol{C}\boldsymbol{V}\tilde{x}$$

After multiplying the right side SSM by $\boldsymbol{V}$, the two SSMs become identical with $x = \boldsymbol{V}\tilde{x}$. Therefore these compute the exact same operator $u \mapsto y$, but with a change of basis by $\boldsymbol{V}$ in the state $x$. $\quad\square$

Lemma 3.1 motivates putting $\boldsymbol{A}$ into a canonical form by conjugation[2], which is ideally more structured and allows faster computation. For example, if $\boldsymbol{A}$ were diagonal, the resulting computations become much more tractable. In particular, the desired $\overline{\boldsymbol{K}}$ (equation (4)) would be a **Vandermonde product** which theoretically only needs $O((N + L)\log^2(N + L))$ arithmetic operations (Pan, 2001).

Unfortunately, the naive application of diagonalization does not work due to numerical issues. First, Vandermonde multiplication is itself a famously ill-conditioned problem (Pan, 2016). Furthermore, we derive the explicit diagonalization for the HiPPO matrix (2) and show it has entries exponentially large in the state size $N$, rendering the diagonalization numerically infeasible (e.g. $\boldsymbol{C}\boldsymbol{V}$ in Lemma 3.1 would not be computable). We note that Gu et al. (2021) proposed a different (unimplemented) algorithm to compute $\overline{\boldsymbol{K}}$ faster than the naive algorithm. In Appendix B, we prove that it is also numerically unstable for related reasons.

**Lemma 3.2.** *The HiPPO matrix $\boldsymbol{A}$ in equation* (2) *is diagonalized by the matrix* $\boldsymbol{V}_{ij} = \binom{i+j}{i-j}$. *In particular,* $\boldsymbol{V}_{3i,i} = \binom{4i}{2i} \approx 2^{4i}$. *Therefore $\boldsymbol{V}$ has entries of magnitude up to* $2^{4N/3}$.

## 3.2 THE S4 PARAMETERIZATION: NORMAL PLUS LOW-RANK

The previous discussion implies that we should only conjugate by well-conditioned matrices $\boldsymbol{V}$. The ideal scenario is when the matrix $\boldsymbol{A}$ is diagonalizable by a perfectly conditioned (i.e., unitary) matrix. By the Spectral Theorem of linear algebra, this is exactly the class of **normal matrices**. However, this class of matrices is restrictive; in particular, it does not contain the HiPPO matrix (2).

We make the observation that although the HiPPO matrix is not normal, it can be decomposed as the *sum of a normal and low-rank matrix*. However, this is still not useful by itself: unlike a diagonal matrix, powering up this sum (in (5)) is still slow and not easily optimized. We overcome this bottleneck by simultaneously applying three new techniques.

- Instead of computing $\overline{\boldsymbol{K}}$ directly, we compute its spectrum by evaluating its **truncated generating function** $\sum_{j=0}^{L-1} \overline{\boldsymbol{K}}_j \zeta^j$ at the roots of unity $\zeta$. $\overline{\boldsymbol{K}}$ can then be found by applying an inverse FFT.

- This generating function is closely related to the matrix resolvent, and now involves a matrix *inverse* instead of *power*. The low-rank term can now be corrected by applying the **Woodbury identity** which reduces $(\boldsymbol{A} + \boldsymbol{P}\boldsymbol{Q}^*)^{-1}$ in terms of $\boldsymbol{A}^{-1}$, truly reducing to the diagonal case.

- Finally, we show that the diagonal matrix case is equivalent to the computation of a **Cauchy kernel** $\frac{1}{\omega_j - \zeta_k}$, a well-studied problem with stable near-linear algorithms (Pan, 2015; 2017).

---

[2]Note that although we ultimately require $\overline{\boldsymbol{A}}$, conjugation commutes with discretization so we refer to $\boldsymbol{A}$.

Our techniques apply to any matrix that can be decomposed as ***Normal Plus Low-Rank (NPLR)***.

**Theorem 1.** *All HiPPO matrices from (Gu et al., 2020a) have a NPLR representation*

$$\boldsymbol{A} = \boldsymbol{V}\boldsymbol{\Lambda}\boldsymbol{V}^* - \boldsymbol{P}\boldsymbol{Q}^\top = \boldsymbol{V}\left(\boldsymbol{\Lambda} - (\boldsymbol{V}^*\boldsymbol{P})(\boldsymbol{V}^*\boldsymbol{Q})^*\right)\boldsymbol{V}^* \tag{6}$$

*for unitary $\boldsymbol{V} \in \mathbb{C}^{N \times N}$, diagonal $\boldsymbol{\Lambda}$, and low-rank factorization $\boldsymbol{P}, \boldsymbol{Q} \in \mathbb{R}^{N \times r}$. These matrices HiPPO- LegS, LegT, LagT all satisfy $r = 1$ or $r = 2$. In particular, equation (2) is NPLR with $r = 1$.*

### 3.3 S4 Algorithms and Computational Complexity

By equation (6), note that NPLR matrices can be conjugated into *diagonal plus low-rank* (DPLR) form (now over $\mathbb{C}$ instead of $\mathbb{R}$). Theorems 2 and 3 describe the complexities of SSMs where $\boldsymbol{A}$ is in DPLR form. S4 is optimal or near-optimal for both recurrent and convolutional representations.

**Theorem 2** (S4 Recurrence). *Given any step size $\Delta$, computing one step of the recurrence (3) can be done in $O(N)$ operations where $N$ is the state size.*

Theorem 2 follows from the fact that the inverse of a DPLR matrix is also DPLR (e.g. also by the Woodbury identity). This implies that the discretized matrix $\overline{\boldsymbol{A}}$ is the product of two DPLR matrices and thus has $O(N)$ matrix-vector multiplication. Appendix C.2 computes $\overline{\boldsymbol{A}}$ in closed DPLR form.

**Theorem 3** (S4 Convolution). *Given any step size $\Delta$, computing the SSM convolution filter $\overline{\boldsymbol{K}}$ can be reduced to 4 Cauchy multiplies, requiring only $\widetilde{O}(N + L)$ operations and $O(N + L)$ space.*

Appendix C, Definition 3 formally defines Cauchy matrices, which are related to rational interpolation problems. Computing with Cauchy matrices is an extremely well-studied problem in numerical analysis, with both fast arithmetic and numerical algorithms based on the famous Fast Multipole Method (FMM) (Pan, 2001; 2015; 2017). The computational complexities of these algorithms under various settings are described in Appendix C, Proposition 5.

We reiterate that Theorem 3 is our core technical contribution, and its algorithm is the very motivation of the NPLR S4 parameterization. This algorithm is formally sketched in Algorithm 1.

### 3.4 Architecture Details of the Deep S4 Layer

Concretely, an S4 layer is parameterized as follows. First initialize a SSM with $\boldsymbol{A}$ set to the HiPPO matrix (2). By Lemma 3.1 and Theorem 1, this SSM is unitarily equivalent to some $(\boldsymbol{\Lambda} - \boldsymbol{P}\boldsymbol{Q}^*, \boldsymbol{B}, \boldsymbol{C})$ for some diagonal $\boldsymbol{\Lambda}$ and vectors $\boldsymbol{P}, \boldsymbol{Q}, \boldsymbol{B}, \boldsymbol{C} \in \mathbb{C}^{N \times 1}$. These comprise S4's $5N$ trainable parameters.

The overall deep neural network (DNN) architecture of S4 is similar to prior work. As defined above, S4 defines a map from $\mathbb{R}^L \to \mathbb{R}^L$, i.e. a 1-D sequence map. Typically, DNNs operate on feature maps of size $H$ instead of 1. S4 handles multiple features by simply defining $H$ independent copies of itself, and then mixing the $H$ features with a position-wise linear layer for a total of $O(H^2) + O(HN)$ parameters per layer. Nonlinear activation functions are also inserted between these layers. Overall, S4 defines a sequence-to-sequence map of shape (batch size, sequence length, hidden dimension), exactly the same as related sequence models such as Transformers, RNNs, and CNNs.

## 4 Experiments

Section 4.1 benchmarks S4 against the LSSL and efficient Transformer models. Section 4.2 validates S4 on LRDs: the LRA benchmark and raw speech classification. Section 4.3 investigates whether S4 can be used as a general sequence model to perform effectively and efficiently in a wide variety of settings including image classification, image and text generation, and time series forecasting.

### 4.1 S4 Efficiency Benchmarks

We benchmark that S4 can be trained quickly and efficiently, both compared to the LSSL, as well as efficient Transformer variants designed for long-range sequence modeling. As outlined in Section 3, S4 is theoretically much more efficient than the LSSL, and Table 1 confirms that the S4 is orders of magnitude more speed- and memory-efficient for practical layer sizes. In fact, S4's speed and memory use is competitive with the most efficient Transformer variants benchmarked by Tay et al. (2021)—Linear Transformer (Katharopoulos et al., 2020) and Performer (Choromanski et al., 2020)—in a parameter-matched setting (Table 2, following the protocol of Tay et al. (2021)).

Table 1: Deep SSMs: The S4 parameterization with Algorithm 1 is asymptotically more efficient than the LSSL.

| | TRAINING STEP (MS) | | | MEMORY ALLOC. (MB) | | |
|---|---|---|---|---|---|---|
| Dim. | 128 | 256 | 512 | 128 | 256 | 512 |
| LSSL | 9.32 | 20.6 | 140.7 | 222.1 | 1685 | 13140 |
| **S4** | 4.77 | 3.07 | 4.75 | 5.3 | 12.6 | 33.5 |
| Ratio | 1.9× | 6.7× | **29.6×** | 42.0× | 133× | **392×** |

Table 2: Benchmarks vs. efficient Transformers

| | LENGTH 1024 | | LENGTH 4096 | |
|---|---|---|---|---|
| | Speed | Mem. | Speed | Mem. |
| Transformer | 1× | 1× | 1× | 1× |
| Performer | 1.23× | 0.43× | 3.79× | 0.086× |
| Linear Trans. | **1.58×** | **0.37×** | 5.35× | **0.067×** |
| **S4** | **1.58×** | 0.43× | 5.19× | 0.091× |

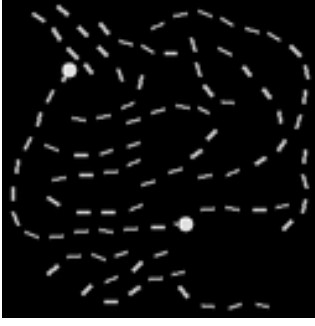 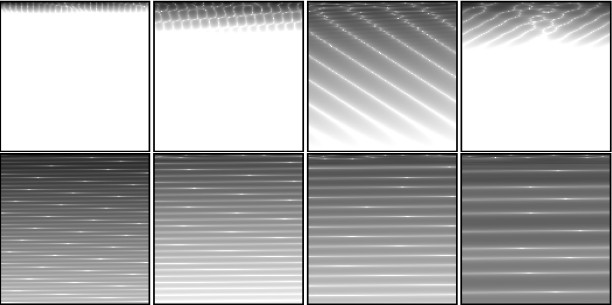

Figure 2: Visualizations of a trained S4 model on LRA Path-X. SSM convolution kernels $\overline{K} \in \mathbb{R}^{16384}$ are reshaped into a $128 \times 128$ image. (*Left*) Example from the Path-X task, which involves deducing if the markers are connected by a path (*Top*) Filters from the first layer (*Bottom*) Filters from the last layer.

Table 3: (**Long Range Arena**) Accuracy on full suite of LRA tasks. (*Top*) Original Transformer variants in LRA. Full results in Appendix D.2. (*Bottom*) Other models reported in the literature.

| MODEL | LISTOPS | TEXT | RETRIEVAL | IMAGE | PATHFINDER | PATH-X | AVG |
|---|---|---|---|---|---|---|---|
| Transformer | 36.37 | 64.27 | 57.46 | 42.44 | 71.40 | ✗ | 53.66 |
| Reformer | 37.27 | 56.10 | 53.40 | 38.07 | 68.50 | ✗ | 50.56 |
| BigBird | 36.05 | 64.02 | 59.29 | 40.83 | 74.87 | ✗ | 54.17 |
| Linear Trans. | 16.13 | 65.90 | 53.09 | 42.34 | 75.30 | ✗ | 50.46 |
| Performer | 18.01 | 65.40 | 53.82 | 42.77 | 77.05 | ✗ | 51.18 |
| FNet | 35.33 | 65.11 | 59.61 | 38.67 | 77.80 | ✗ | 54.42 |
| Nyströmformer | 37.15 | 65.52 | 79.56 | 41.58 | 70.94 | ✗ | 57.46 |
| Luna-256 | 37.25 | 64.57 | 79.29 | 47.38 | 77.72 | ✗ | 59.37 |
| **S4** | **58.35** | **76.02** | **87.09** | **87.26** | **86.05** | **88.10** | **80.48** |

## 4.2 LEARNING LONG RANGE DEPENDENCIES

As described in Sections 2.2 and 3.1, S4 uses a principled approach to address LRDs based on the HiPPO theory of continuous-time memorization. Our goal in this section is to validate that S4 achieves high performance on difficult tasks that require long-range reasoning. We focus here on two problems: (i) the Long-Range Arena, a well-known benchmark designed to test efficient sequence models on LRDs, and (ii) a speech classification problem as a real-world test of LRDs.

**Long Range Arena (LRA).** LRA (Tay et al., 2021) contains 6 tasks with lengths 1K-16K steps, encompassing modalities and objectives that require similarity, structural, and visuospatial reasoning. Table 3 compares S4 against the 11 Transformer variants from Tay et al. (2021) as well as follow-up work. S4 substantially advances the SoTA, outperforming all baselines on all tasks and averaging $80.48\%$ compared to less than $60\%$ for every baseline. Notably, S4 solves the Path-X task, an extremely challenging task that involves reasoning about LRDs over sequences of length $128 \times 128 = 16384$. All previous models have failed (i.e. random guessing) due to memory or computation bottlenecks, or simply being unable to learn such long dependencies.

We analyze S4's performance on Path-X by visualizing its learned representations, in particular 1-D convolution kernels $\overline{K}$ which are the focus of our technical results in Section 3. Fig. 2 shows that S4

learns a variety of filters that display spatially consistent structure and demonstrate awareness of the 2-D nature of the data. In particular, the lower layers learn simple kernels that extract features from just a few rows of local context while ignoring the rest of the image. On the other hand, higher layers aggregate information globally across full columns of the image at varying spatial frequencies. Filters in these higher layers span the entire context (16384 pixels), confirming S4's ability to learn LRDs.

**Raw Speech Classification.** Speech is a typical real-world time series domain, involving signals sampled from an underlying physical process at high frequency. We perform speech classification using the *Speech Commands* dataset (Warden, 2018). While most sequence models for speech rely on extensive preprocessing (e.g. to MFCC features), we classify raw speech (length-16000) following Romero et al. (2021). S4 achieves 98.3% accuracy, higher than all baselines that use the $100\times$ shorter MFCC features, and validates that a powerful LRD model is able to extract more information from the raw data and outperform hand-crafted pre-processing. Additionally, we include a baseline CNN specifically designed for raw speech, the discriminator from the WaveGAN model (Donahue et al., 2019), which performs worse than S4 while having $90\times$ more parameters and incorporating many more architectural heuristics (Appendix D.2).

### 4.3 S4 AS A GENERAL SEQUENCE MODEL

A key goal of sequence modeling research is to develop a single model that can be applied in many domains (e.g. images, audio, text, time-series) with a broad range of capabilities (e.g. efficient training, fast generation, handling irregularly sampled data). As a fundamental scientific model, SSMs are a promising candidate that come with a range of capabilities, and S4's strong results on LRD benchmarks spanning images, text, and speech are evidence of S4's potential as a general sequence model. In this section, we focus on understanding this question in more depth by highlighting key strengths of S4 in settings that usually require specialized models. The tasks we focus on (generative modeling, image classification, time-series forecasting) are considered as LRD tasks in the literature, and serve as additional validation that S4 handles LRDs efficiently.

**Large-scale generative modeling.** We investigate two well-studied image and text benchmarks to validate the scalability, flexibility, and efficiency of S4. These tasks require much larger models than our previous tasks – up to 250M parameters.

First, CIFAR density estimation is a popular benchmark for autoregressive models, where images are flattened into a sequence of 3072 RGB subpixels that are predicted one by one. Table 6 shows that *with no 2D inductive bias*, S4 is competitive with the best models designed for this task.

Second, WikiText-103 is an established benchmark for language modeling, an important task for large-scale sequence models where tokens are predicted sequentially based on past context. Although RNNs were the model of choice for many years, Transformers are now the dominant model in such applications that contain data that is inherently discrete. We show that alternative models

Table 4: (**Speech classification**) Transformer, CTM, RNN, CNN, and SSM models. (*MFCC*) Standard pre-processed MFCC features (length-161). (*Raw*) Unprocessed signals (length-16000). (*0.5×*) Frequency change at test time. ✗ denotes not applicable or computationally infeasible on single GPU.

|  | MFCC | RAW | 0.5× |
|---|---|---|---|
| Transformer | 90.75 | ✗ | ✗ |
| Performer | 80.85 | 30.77 | 30.68 |
| ODE-RNN | 65.9 | ✗ | ✗ |
| NRDE | 89.8 | 16.49 | 15.12 |
| ExpRNN | 82.13 | 11.6 | 10.8 |
| LipschitzRNN | 88.38 | ✗ | ✗ |
| CKConv | **95.3** | 71.66 | 65.96 |
| WaveGAN-D | ✗ | 96.25 | ✗ |
| LSSL | 93.58 | ✗ | ✗ |
| **S4** | 93.96 | **98.32** | **96.30** |

Table 5: (**Pixel-level 1-D image classification**) Transformer, RNN, CNN, and SSM models. Extended results + citations in Appendix D.

|  | sMNIST | pMNIST | sCIFAR |
|---|---|---|---|
| Transformer | 98.9 | 97.9 | 62.2 |
| LSTM | 98.9 | 95.11 | 63.01 |
| r-LSTM | 98.4 | 95.2 | 72.2 |
| UR-LSTM | 99.28 | 96.96 | 71.00 |
| UR-GRU | 99.27 | 96.51 | 74.4 |
| HiPPO-RNN | 98.9 | 98.3 | 61.1 |
| LMU-FFT | - | 98.49 | - |
| LipschitzRNN | 99.4 | 96.3 | 64.2 |
| TCN | 99.0 | 97.2 | - |
| TrellisNet | 99.20 | 98.13 | 73.42 |
| CKConv | 99.32 | 98.54 | 63.74 |
| LSSL | 99.53 | **98.76** | 84.65 |
| **S4** | **99.63** | 98.70 | **91.13** |

Table 6: (**CIFAR-10 density estimation**) As a generic sequence model, S4 is competitive with previous autoregressive models (in bits per dim.) while incorporating no 2D inductive bias, and has fast generation through its recurrence mode.

| Model | bpd | 2D bias | Images / sec |
|---|---|---|---|
| Transformer | 3.47 | **None** | 0.32 (1×) |
| Linear Transf. | 3.40 | **None** | 17.85 (56×) |
| PixelCNN | 3.14 | 2D conv. | - |
| Row PixelRNN | 3.00 | 2D BiLSTM | - |
| PixelCNN++ | 2.92 | 2D conv. | 19.19 (59.97×) |
| Image Transf. | 2.90 | 2D local attn. | 0.54 (1.7×) |
| PixelSNAIL | 2.85 | 2D conv. + attn. | 0.13 (0.4×) |
| Sparse Transf. | **2.80** | 2D sparse attn. | - |
| **S4** (base) | 2.92 | **None** | **20.84 (65.1×)** |
| **S4** (large) | 2.85 | **None** | 3.36 (10.5×) |

Table 7: (**WikiText-103 language modeling**) S4 approaches the performance of Transformers with much faster generation. (*Top*) Transformer baseline which our implementation is based on, with attention replaced by S4. (*Bottom*) Attention-free models (RNNs and CNNs).

| Model | Params | Test ppl. | Tokens / sec |
|---|---|---|---|
| Transformer | 247M | **20.51** | 0.8K (1×) |
| GLU CNN | 229M | 37.2 | - |
| AWD-QRNN | 151M | 33.0 | - |
| LSTM + Hebb. | - | 29.2 | - |
| TrellisNet | 180M | 29.19 | - |
| Dynamic Conv. | 255M | 25.0 | - |
| TaLK Conv. | 240M | 23.3 | - |
| **S4** | 249M | **21.28** | **48K (60×)** |

to Transformers can still be competitive in these settings. By simply taking a strong Transformer baseline (Baevski & Auli, 2018) and replacing the self-attention layers, S4 substantially closes the gap to Transformers (within 0.8 ppl), setting SoTA for attention-free models by over 2 ppl.

**Fast autoregressive inference.** A prominent limitation of autoregressive models is inference speed (e.g. generation), since they require a pass over the full context for every new sample. Several methods have been specifically crafted to overcome this limitation such as the Linear Transformer, a hybrid Transformer/RNN that switches to a stateful, recurrent view at inference time for speed.

As a stateful model, SSMs automatically have this ability (Fig. 1). By switching to its recurrent representation (Section 2.3), S4 requires *constant memory and computation* per time step – in contrast to standard autoregressive models which scale in the context length. On both CIFAR-10 and WikiText-103, we report the throughput of various models at generation time, with S4 around 60× faster than a vanilla Transformer on both tasks (details in Appendix D.3.3).

**Sampling resolution change.** As a continuous-time model, S4 automatically adapts to data sampled at different rates, a challenging setting for time series with a dedicated line of work (Rubanova et al., 2019; De Brouwer et al., 2019; Romero et al., 2021). Without re-training, S4 achieves 96.3% accuracy at 0.5× the frequency on Speech Commands (Table 4), simply by changing its internal step size $\Delta$ (Section 2.3).

**Learning with weaker inductive bias.** Beyond our results on speech (Section 4.2), we further validate that S4 can be applied with minimal modifications on two domains that typically require specialized domain-specific preprocessing and architectures. First, we compare S4 to the Informer (Zhou et al., 2021), a new Transformer architecture that uses a complex encoder-decoder designed for time-series forecasting problems. A simple application of S4 that treats forecasting as a masked sequence-to-sequence transformation (Fig. 3) outperforms the Informer and other baselines on 40/50 settings across 5 forecasting tasks. Notably, S4 is better on the longest setting in each task, e.g. reducing MSE by 37% when forecasting 30 days of weather data (Appendix D.3.5).

Finally, we evaluate S4 on pixel-level sequential image classification tasks (Table 5), popular benchmarks which were originally LRD tests for RNNs (Arjovsky et al., 2016). Beyond LRDs, these benchmarks point to a recent effort of the ML community to solve vision problems with reduced domain knowledge, in the spirit of models such as Vision Transformers (Dosovitskiy et al., 2020) and MLP-Mixer (Tolstikhin et al., 2021) . Sequential CIFAR is a particularly challenging dataset where outside of SSMs, all sequence models have a gap of over 25% to a simple 2-D CNN. By contrast, S4 is competitive with a larger ResNet18 (7.9M vs. 11.0M parameters), both with (**93.16%** vs. 95.62%) or without (**91.12%** vs. 89.46%) data augmentation. Moreover, it is much more robust to other architectural choices (e.g. **90.46%** vs. 79.52% when swapping BatchNorm for LayerNorm).

## 5  CONCLUSION

We introduce S4, a sequence model that uses a new parameterization for the state space model's continuous-time, recurrent, and convolutional views to efficiently model LRDs in a principled manner. Results across established benchmarks evaluating a diverse range of data modalities and model capabilities suggest that S4 has the potential to be an effective general sequence modeling solution.

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

## A  DISCUSSION

**Related Work.** Our work is most closely related to a line of work originally motivated by a particular biologically-inspired SSM, which led to mathematical models for addressing LRDs. Voelker (2019); Voelker et al. (2019) derived a non-trainable SSM motivated from approximating a neuromorphic spiking model, and Chilkuri & Eliasmith (2021) showed that it could be sped up at train time with a convolutional view. Gu et al. (2020a) extended this special case to a general continuous-time function approximation framework with several more special cases of $A$ matrices designed for long-range dependencies. However, instead of using a true SSM, all of these works fixed a choice of $A$ and built RNNs around it. Most recently, Gu et al. (2021) used the full (1) explicitly as a deep SSM model, exploring new conceptual views of SSMs, as well as allowing $A$ to be trained. As mentioned in Section 1, their method used a naive instantiation of SSMs that suffered from an additional factor of $N$ in memory and $N^2$ in computation.

Beyond this work, our technical contributions (Section 3) on the S4 parameterization and algorithms are applicable to a broader family of SSMs including these investigated in prior works, and our techniques for working with these models may be of independent interest.

**Implementation.** The computational core of S4's training algorithm is the Cauchy kernel discussed in Sections 3.2 and 3.3 and Appendix C.3. As described in Appendix C.3 Proposition 5, there are many algorithms for it with differing computational complexities and sophistication. Our current implementation of S4 actually uses the naive $O(NL)$ algorithm which is easily parallelized on GPUs and has more easily accessible libraries allowing it to be implemented; we leverage the `pykeops` library for memory-efficient kernel operations. However, this library is a much more general library that may not be optimized for the Cauchy kernels used here, and we believe that a dedicated CUDA implementation can be more efficient. Additionally, as discussed in this work, there are asymptotically faster and numerically stable algorithms for the Cauchy kernel (Proposition 5). However, these algorithms are currently not implemented for GPUs due to a lack of previous applications that require them. We believe that more efficient implementations of these self-contained computational kernels

are possible, and that S4 (and SSMs at large) may have significant room for further improvements in efficiency.

**Limitations and Future Directions.** In this work, we show that S4 can address a wide variety of data effectively. However, it may not necessarily be the most suitable model for all types of data. For example, Table 7 still found a gap compared to Transformers for language modeling. An interesting future direction is exploring combinations of S4 with other sequence models to complement their strengths. We are excited about other directions, including continuing to explore the benefits of S4 on audio data (e.g. pre-training or generation settings), and generalizing HiPPO and S4 to higher-dimensional data for image and video applications.

# B NUMERICAL INSTABILITY OF LSSL

This section proves the claims made in Section 3.1 about prior work. We first derive the explicit diagonalization of the HiPPO matrix, confirming its instability because of exponentially large entries. We then discuss the proposed theoretically fast algorithm from (Gu et al., 2021) (Theorem 2) and show that it also involves exponentially large terms and thus cannot be implemented.

## B.1 HiPPO DIAGONALIZATION

*Proof of Lemma 3.2.* The HiPPO matrix (2) is equal, up to sign and conjugation by a diagonal matrix, to

$$
\boldsymbol{A} = \begin{bmatrix}
1 & & & & & & & \\
-1 & 2 & & & & & & \\
1 & -3 & 3 & & & & & \\
-1 & 3 & -5 & 4 & & & & \\
1 & -3 & 5 & -7 & 5 & & & \\
-1 & 3 & -5 & 7 & -9 & 6 & & \\
1 & -3 & 5 & -7 & 9 & -11 & 7 & \\
-1 & 3 & -5 & 7 & -9 & 11 & -13 & 8 \\
\vdots & & & & & & & & \ddots
\end{bmatrix}
$$

$$
\boldsymbol{A}_{nk} = \begin{cases}
(-1)^{n-k}(2k+1) & n > k \\
k+1 & n = k \\
0 & n < k
\end{cases}.
$$

Our goal is to show that this $\boldsymbol{A}$ is diagonalized by the matrix

$$
\boldsymbol{V} = \binom{i+j}{i-j}_{ij} = \begin{bmatrix}
1 & & & & & \\
1 & 1 & & & & \\
1 & 3 & 1 & & & \\
1 & 6 & 5 & 1 & & \\
1 & 10 & 15 & 7 & 1 & \\
1 & 15 & 35 & 28 & 9 & 1 \\
\vdots & & & & & & \ddots
\end{bmatrix},
$$

or in other words that columns of this matrix are eigenvectors of $\boldsymbol{A}$.

Concretely, we will show that the $j$-th column of this matrix $\boldsymbol{v}^{(j)}$ with elements

$$
\boldsymbol{v}_i^{(j)} = \begin{cases}
0 & i < j \\
\binom{i+j}{i-j} = \binom{i+j}{2j} & i \geq j
\end{cases}
$$

is an eigenvector with eigenvalue $j + 1$. In other words we must show that for all indices $k \in [N]$,

$$
(\boldsymbol{A}\boldsymbol{v}^{(j)})_k = \sum_i \boldsymbol{A}_{ki}\boldsymbol{v}_i^{(j)} = (j+1)\boldsymbol{v}_k^{(j)}. \tag{7}
$$

If $k < j$, then for all $i$ inside the sum, either $k < i$ or $i < j$. In the first case $\boldsymbol{A}_{ki} = 0$ and in the second case $\boldsymbol{v}_i^{(j)} = 0$, so both sides of equation (7) are equal to 0.

It remains to show the case $k \geq j$, which proceeds by induction on $k$. Expanding equation (7) using the formula for $\boldsymbol{A}$ yields

$$(\boldsymbol{A}\boldsymbol{v})_k^{(j)} = \sum_i \boldsymbol{A}_{ki}\boldsymbol{v}_i^{(j)} = \sum_{i=j}^{k-1}(-1)^{k-i}(2i+1)\binom{i+j}{2j} + (k+1)\binom{k+j}{2j}.$$

In the base case $k = j$, the sum disappears and we are left with $(\boldsymbol{A}\boldsymbol{v}^{(j)})_j = (j+1)\binom{2j}{2j} = (j+1)\boldsymbol{v}_j^{(j)}$, as desired.

Otherwise, the sum for $(\boldsymbol{A}\boldsymbol{v})_k^{(j)}$ is the same as the sum for $(\boldsymbol{A}\boldsymbol{v})_{k-1}^{(j)}$ but with sign reversed and a few edge terms. The result follows from applying the inductive hypothesis and algebraic simplification:

$$\begin{aligned}
(\boldsymbol{A}\boldsymbol{v})_k^{(j)} &= -(\boldsymbol{A}\boldsymbol{v})_{k-1}^{(j)} - (2k-1)\binom{k-1+j}{2j} + k\binom{k-1+j}{2j} + (k+1)\binom{k+j}{2j} \\
&= -(j+1)\binom{k-1+j}{2j} - (k-1)\binom{k-1+j}{2j} + (k+1)\binom{k+j}{2j} \\
&= -(j+k)\binom{k-1+j}{2j} + (k+1)\binom{k+j}{2j} \\
&= -(j+k)\frac{(k-1+j)!}{(k-1-j)!(2j)!} + (k+1)\binom{k+j}{2j} \\
&= -\frac{(k+j)!}{(k-1-j)!(2j)!} + (k+1)\binom{k+j}{2j} \\
&= -(k-j)\frac{(k+j)!}{(k-j)!(2j)!} + (k+1)\binom{k+j}{2j} \\
&= (j-k)(k+1)\binom{k+j}{2j} + (k+1)\binom{k+j}{2j} \\
&= (j+1)\boldsymbol{v}_k^{(j)}.
\end{aligned}$$

$\square$

## B.2 Fast but Unstable LSSL Algorithm

Instead of diagonalization, Gu et al. (2021, Theorem 2) proposed a sophisticated fast algorithm to compute

$$K_L(\overline{\boldsymbol{A}}, \overline{\boldsymbol{B}}, \overline{\boldsymbol{C}}) = (\overline{\boldsymbol{C}\boldsymbol{B}}, \overline{\boldsymbol{C}\boldsymbol{A}\boldsymbol{B}}, \ldots, \overline{\boldsymbol{C}\boldsymbol{A}}^{L-1}\overline{\boldsymbol{B}}).$$

This algorithm runs in $O(N \log^2 N + L \log L)$ operations and $O(N + L)$ space. However, we now show that this algorithm is also numerically unstable.

There are several reasons for the instability of this algorithm, but most directly we can pinpoint a particular intermediate quantity that they use.

**Definition 1.** *The fast LSSL algorithm computes coefficients of $p(x)$, the characteristic polynomial of $A$, as an intermediate computation. Additionally, it computes the coefficients of its inverse, $p(x)^{-1}$ (mod $x^L$).*

We now claim that this quantity is numerically unfeasible. We narrow down to the case when $\overline{\boldsymbol{A}} = \boldsymbol{I}$ is the identity matrix. Note that this case is actually in some sense the most typical case: when discretizing the continuous-time SSM to discrete-time by a step-size $\Delta$, the discretized transition matrix $\overline{\boldsymbol{A}}$ is brought closer to the identity. For example, with the Euler discretization $\overline{\boldsymbol{A}} = \boldsymbol{I} + \Delta\boldsymbol{A}$, we have $\overline{\boldsymbol{A}} \to \boldsymbol{I}$ as the step size $\Delta \to 0$.

**Lemma B.1.** *When $\overline{\boldsymbol{A}} = \boldsymbol{I}$, the fast LSSL algorithm requires computing terms exponentially large in $N$.*

*Proof.* The characteristic polynomial of $\boldsymbol{I}$ is

$$p(x) = \det|\boldsymbol{I} - x\boldsymbol{I}| = (1-x)^N.$$

These coefficients have size up to $\binom{N}{\frac{N}{2}} \approx \frac{2^N}{\sqrt{\pi N/2}}$.

The inverse of $p(x)$ has even larger coefficients. It can be calculated in closed form by the generalized binomial formula:

$$(1-x)^{-N} = \sum_{k=0}^{\infty} \binom{N+k-1}{k} x^k.$$

Taking this $\pmod{x^L}$, the largest coefficient is

$$\binom{N+L-2}{L-1} = \binom{N+L-2}{N-1} = \frac{(L-1)(L-2)\dots(L-N+1)}{(N-1)!}.$$

When $L = N - 1$ this is

$$\binom{2(N-1)}{N-1} \approx \frac{2^{2N}}{\sqrt{\pi N}}$$

already larger than the coefficients of $(1-x)^N$, and only increases as $L$ grows. $\qquad\square$

## C  S4 Algorithm Details

This section proves the results of Section 3.3, providing complete details of our efficient algorithms for S4.

Appendices C.1 to C.3 prove Theorems 1 to 3 respectively.

### C.1  NPLR Representations of HiPPO Matrices

We first prove Theorem 1, showing that all HiPPO matrices for continuous-time memory fall under the S4 normal plus low-rank (NPLR) representation.

*Proof of Theorem 1.* We consider each of the three cases HiPPO-LagT, HiPPO-LegT, and HiPPO-LegS separately. Note that the primary HiPPO matrix defined in this work (equation (2)) is the HiPPO-LegT matrix.

**HiPPO-LagT.** The HiPPO-LagT matrix is simply

$$\boldsymbol{A}_{nk} = \begin{cases} 0 & n < k \\ -\frac{1}{2} & n = k \\ -1 & n > k \end{cases}$$

$$\boldsymbol{A} = - \begin{bmatrix} \frac{1}{2} & & & & \cdots \\ 1 & \frac{1}{2} & & & \\ 1 & 1 & \frac{1}{2} & & \\ 1 & 1 & 1 & \frac{1}{2} & \\ \vdots & & & & \ddots \end{bmatrix}.$$

Adding the matrix of all $\frac{1}{2}$, which is rank 1, yields

$$- \begin{bmatrix} & -\frac{1}{2} & -\frac{1}{2} & -\frac{1}{2} \\ \frac{1}{2} & & -\frac{1}{2} & -\frac{1}{2} \\ \frac{1}{2} & \frac{1}{2} & & -\frac{1}{2} \\ \frac{1}{2} & \frac{1}{2} & \frac{1}{2} & \end{bmatrix}.$$

This matrix is now skew-symmetric. Skew-symmetric matrices are a particular case of normal matrices with pure-imaginary eigenvalues.

Gu et al. (2020a) also consider a case of HiPPO corresponding to the generalized Laguerre polynomials that generalizes the above HiPPO-LagT case. In this case, the matrix $\boldsymbol{A}$ (up to conjugation by a diagonal matrix) ends up being close to the above matrix, but with a different element on the diagonal. After adding the rank-1 correction, it becomes the above skew-symmetric matrix plus a multiple of

the identity. Thus after diagonalization by the same matrix as in the LagT case, it is still reduced to diagonal plus low-rank (DPLR) form, where the diagonal is now pure imaginary plus a real constant.

**HiPPO-LegS.** We restate the formula from equation (2) for convenience.

$$A_{nk} = - \begin{cases} (2n+1)^{1/2}(2k+1)^{1/2} & \text{if } n > k \\ n+1 & \text{if } n = k \\ 0 & \text{if } n < k \end{cases}.$$

Adding $\frac{1}{2}(2n+1)^{1/2}(2k+1)^{1/2}$ to the whole matrix gives

$$- \begin{cases} \frac{1}{2}(2n+1)^{1/2}(2k+1)^{1/2} & \text{if } n > k \\ \frac{1}{2} & \text{if } n = k \\ -\frac{1}{2}(2n+1)^{1/2}(2k+1)^{1/2} & \text{if } n < k \end{cases}$$

Note that this matrix is not skew-symmetric, but is $\frac{1}{2}I + S$ where $S$ is a skew-symmetric matrix. This is diagonalizable by the same unitary matrix that diagonalizes $S$.

**HiPPO-LegT.**

Up to the diagonal scaling, the LegT matrix is

$$A = - \begin{bmatrix} 1 & -1 & 1 & -1 & \dots \\ 1 & 1 & -1 & 1 \\ 1 & 1 & 1 & -1 \\ 1 & 1 & 1 & 1 \\ \vdots & & & & \ddots \end{bmatrix}.$$

By adding $-1$ to this matrix and then the matrix

$$\begin{bmatrix} 2 & & 2 & \\ & & & \\ 2 & & 2 & \end{bmatrix}$$

the matrix becomes

$$\begin{bmatrix} & -2 & & -2 \\ 2 & & & \\ & & & -2 \\ 2 & & 2 & \end{bmatrix}$$

which is skew-symmetric. In fact, this matrix is the inverse of the Chebyshev Jacobi.

An alternative way to see this is as follows. The LegT matrix is the inverse of the matrix

$$\begin{bmatrix} -1 & 1 & & 0 \\ -1 & & 1 & \\ & -1 & & 1 \\ & & -1 & -1 \end{bmatrix}$$

This can obviously be converted to a skew-symmetric matrix by adding a rank 2 term. The inverses of these matrices are also rank-2 differences from each other by the Woodbury identity.

A final form is

$$\begin{bmatrix} -1 & 1 & -1 & 1 \\ -1 & -1 & 1 & -1 \\ -1 & -1 & -1 & 1 \\ -1 & -1 & -1 & -1 \end{bmatrix} + \begin{bmatrix} 1 & 0 & 1 & 0 \\ 0 & 1 & 0 & 1 \\ 1 & 0 & 1 & 0 \\ 0 & 1 & 0 & 1 \end{bmatrix} = \begin{bmatrix} 0 & 1 & 0 & 1 \\ -1 & 0 & 1 & 0 \\ 0 & -1 & 0 & 1 \\ -1 & 0 & -1 & 0 \end{bmatrix}$$

This has the advantage that the rank-2 correction is symmetric (like the others), but the normal skew-symmetric matrix is now 2-quasiseparable instead of 1-quasiseparable.

$\square$

## C.2 COMPUTING THE S4 RECURRENT VIEW

We prove Theorem 2 showing the efficiency of the S4 parameterization for computing one step of the recurrent representation (Section 2.3).

Recall that without loss of generality, we can assume that the state matrix $\boldsymbol{A} = \boldsymbol{\Lambda} - \boldsymbol{P}\boldsymbol{Q}^*$ is diagonal plus low-rank (DPLR), potentially over $\mathbb{C}$. Our goal in this section is to explicitly write out a closed form for the discretized matrix $\overline{\boldsymbol{A}}$.

Recall from equation (3) that

$$\overline{\boldsymbol{A}} = (\boldsymbol{I} - \Delta/2 \cdot \boldsymbol{A})^{-1}(\boldsymbol{I} + \Delta/2 \cdot \boldsymbol{A})$$
$$\overline{\boldsymbol{B}} = (\boldsymbol{I} - \Delta/2 \cdot \boldsymbol{A})^{-1}\Delta\boldsymbol{B}.$$

We first simplify both terms in the definition of $\overline{\boldsymbol{A}}$ independently.

**Forward discretization.** The first term is essentially the Euler discretization motivated in Section 2.3.

$$\boldsymbol{I} + \frac{\Delta}{2}\boldsymbol{A} = \boldsymbol{I} + \frac{\Delta}{2}(\boldsymbol{\Lambda} - \boldsymbol{P}\boldsymbol{Q}^*)$$
$$= \frac{\Delta}{2}\left[\frac{2}{\Delta}\boldsymbol{I} + (\boldsymbol{\Lambda} - \boldsymbol{P}\boldsymbol{Q}^*)\right]$$
$$= \frac{\Delta}{2}\boldsymbol{A_0}$$

where $\boldsymbol{A_0}$ is defined as the term in the final brackets.

**Backward discretization.** The second term is known as the Backward Euler's method. Although this inverse term is normally difficult to deal with, in the DPLR case we can simplify it using Woodbury's Identity (Proposition 4).

$$\left(\boldsymbol{I} - \frac{\Delta}{2}\boldsymbol{A}\right)^{-1} = \left(\boldsymbol{I} - \frac{\Delta}{2}(\boldsymbol{\Lambda} - \boldsymbol{P}\boldsymbol{Q}^*)\right)^{-1}$$
$$= \frac{2}{\Delta}\left[\frac{2}{\Delta} - \boldsymbol{\Lambda} + \boldsymbol{P}\boldsymbol{Q}^*\right]^{-1}$$
$$= \frac{2}{\Delta}\left[\boldsymbol{D} - \boldsymbol{D}\boldsymbol{P}\left(\boldsymbol{I} + \boldsymbol{Q}^*\boldsymbol{D}\boldsymbol{P}\right)^{-1}\boldsymbol{Q}^*\boldsymbol{D}\right]$$
$$= \frac{2}{\Delta}\boldsymbol{A_1}$$

where $\boldsymbol{D} = \left(\frac{2}{\Delta} - \boldsymbol{\Lambda}\right)^{-1}$ and $\boldsymbol{A_1}$ is defined as the term in the final brackets. Note that $(1 + \boldsymbol{Q}^*\boldsymbol{D}\boldsymbol{P})$ is actually a scalar in the case when the low-rank term has rank 1.

**S4 Recurrence.** Finally, the full bilinear discretization can be rewritten in terms of these matrices as

$$\overline{\boldsymbol{A}} = \boldsymbol{A_1}\boldsymbol{A_0}$$
$$\overline{\boldsymbol{B}} = \frac{2}{\Delta}\boldsymbol{A_1}\Delta\boldsymbol{B} = 2\boldsymbol{A_1}\boldsymbol{B}.$$

The discrete-time SSM (3) becomes

$$x_k = \overline{\boldsymbol{A}}x_{k-1} + \overline{\boldsymbol{B}}u_k$$
$$= \boldsymbol{A_1}\boldsymbol{A_0}x_{k-1} + 2\boldsymbol{A_1}\boldsymbol{B}u_k$$
$$y_k = \boldsymbol{C}x_k.$$

Note that $\boldsymbol{A_0}, \boldsymbol{A_1}$ are accessed only through matrix-vector multiplications. Since they are both DPLR, they have $O(N)$ matrix-vector multiplication, showing Theorem 2.

## C.3 COMPUTING THE CONVOLUTIONAL VIEW

The most involved part of using SSMs efficiently is computing $\overline{\boldsymbol{K}}$. This algorithm was sketched in Section 3.2 and is the main motivation for the S4 parameterization. In this section, we define the necessary intermediate quantities and prove the main technical result.

The algorithm for Theorem 3 falls in roughly three stages, leading to Algorithm 1. Assuming $\boldsymbol{A}$ has been conjugated into diagonal plus low-rank form, we successively simplify the problem of computing $\overline{\boldsymbol{K}}$ by applying the techniques outlined in Section 3.2.

**Remark C.1.** *We note that for the remainder of this section, we transpose $\boldsymbol{C}$ to be a column vector of shape $\mathbb{C}^N$ or $\mathbb{C}^{N \times 1}$ instead of matrix or row vector $\mathbb{C}^{1 \times N}$ as in (1). In other words the SSM is*

$$
\begin{aligned}
x'(t) &= \boldsymbol{A}x(t) + \boldsymbol{B}u(t) \\
y(t) &= \boldsymbol{C}^* x(t) + \boldsymbol{D}u(t).
\end{aligned}
\tag{8}
$$

*This convention is made so that $\boldsymbol{C}$ has the same shape as $\boldsymbol{B}, \boldsymbol{P}, \boldsymbol{Q}$ and simplifies the implementation of S4.*

**Reduction 0: Diagonalization**  By Lemma 3.1, we can switch the representation by conjugating with any unitary matrix. For the remainder of this section, we can assume that $\boldsymbol{A}$ is (complex) diagonal plus low-rank (DPLR).

Note that unlike diagonal matrices, a DPLR matrix does not lend itself to efficient computation of $\overline{\boldsymbol{K}}$. The reason is that $\overline{\boldsymbol{K}}$ computes terms $\overline{\boldsymbol{C}}^* \overline{\boldsymbol{A}}^i \overline{\boldsymbol{B}}$ which involve powers of the matrix $\overline{\boldsymbol{A}}$. These are trivially computable when $\overline{\boldsymbol{A}}$ is diagonal, but is no longer possible for even simple modifications to diagonal matrices such as DPLR.

**Reduction 1: SSM Generating Function**  To address the problem of computing powers of $\overline{\boldsymbol{A}}$, we introduce another technique. Instead of computing the SSM convolution filter $\overline{\boldsymbol{K}}$ directly, we introduce a generating function on its coefficients and compute evaluations of it.

**Definition 2** (SSM Generating Function). *We define the following quantities:*

- *The* SSM convolution function *is $\mathcal{K}(\overline{\boldsymbol{A}}, \overline{\boldsymbol{B}}, \overline{\boldsymbol{C}}) = (\overline{\boldsymbol{C}}^* \overline{\boldsymbol{B}}, \overline{\boldsymbol{C}}^* \overline{\boldsymbol{A}} \overline{\boldsymbol{B}}, \dots)$ and the (truncated) SSM filter of length $L$*

$$
\mathcal{K}_L(\overline{\boldsymbol{A}}, \overline{\boldsymbol{B}}, \overline{\boldsymbol{C}}) = (\overline{\boldsymbol{C}}^* \overline{\boldsymbol{B}}, \overline{\boldsymbol{C}}^* \overline{\boldsymbol{A}} \overline{\boldsymbol{B}}, \dots, \overline{\boldsymbol{C}}^* \overline{\boldsymbol{A}}^{L-1} \overline{\boldsymbol{B}}) \in \mathbb{R}^L
\tag{9}
$$

- *The* SSM generating function *at node $z$ is*

$$
\hat{\mathcal{K}}(z; \overline{\boldsymbol{A}}, \overline{\boldsymbol{B}}, \overline{\boldsymbol{C}}) \in \mathbb{C} := \sum_{i=0}^{\infty} \overline{\boldsymbol{C}}^* \overline{\boldsymbol{A}}^i \overline{\boldsymbol{B}} z^i = \overline{\boldsymbol{C}}^* (\boldsymbol{I} - \overline{\boldsymbol{A}} z)^{-1} \overline{\boldsymbol{B}}
\tag{10}
$$

  *and the* truncated SSM generating function *at node $z$ is*

$$
\hat{\mathcal{K}}_L(z; \overline{\boldsymbol{A}}, \overline{\boldsymbol{B}}, \overline{\boldsymbol{C}})^* \in \mathbb{C} := \sum_{i=0}^{L-1} \overline{\boldsymbol{C}}^* \overline{\boldsymbol{A}}^i \overline{\boldsymbol{B}} z^i = \overline{\boldsymbol{C}}^* (\boldsymbol{I} - \overline{\boldsymbol{A}}^L z^L)(\boldsymbol{I} - \overline{\boldsymbol{A}} z)^{-1} \overline{\boldsymbol{B}}
\tag{11}
$$

- *The truncated SSM generating function at nodes $\Omega \in \mathbb{C}^M$ is*

$$
\hat{\mathcal{K}}_L(\Omega; \overline{\boldsymbol{A}}, \overline{\boldsymbol{B}}, \overline{\boldsymbol{C}}) \in \mathbb{C}^M := \left( \hat{\mathcal{K}}_L(\omega_k; \overline{\boldsymbol{A}}, \overline{\boldsymbol{B}}, \overline{\boldsymbol{C}}) \right)_{k \in [M]}
\tag{12}
$$

Intuitively, the generating function essentially converts the SSM convolution filter from the time domain to frequency domain. Importantly, it preserves the same information, and the desired SSM convolution filter can be recovered from evaluations of its generating function.

**Lemma C.2.** *The SSM function $\mathcal{K}_L(\overline{\boldsymbol{A}}, \overline{\boldsymbol{B}}, \overline{\boldsymbol{C}})$ can be computed from the SSM generating function $\hat{\mathcal{K}}_L(\Omega; \overline{\boldsymbol{A}}, \overline{\boldsymbol{B}}, \overline{\boldsymbol{C}})$ at the roots of unity $\Omega = \{\exp(-2\pi i \frac{k}{L} : k \in [L]\}$ stably in $O(L \log L)$ operations.*

*Proof.* For convenience define

$$
\begin{aligned}
\overline{\boldsymbol{K}} &= \mathcal{K}_L(\overline{\boldsymbol{A}}, \overline{\boldsymbol{B}}, \overline{\boldsymbol{C}}) \\
\hat{\boldsymbol{K}} &= \hat{\mathcal{K}}_L(\Omega; \overline{\boldsymbol{A}}, \overline{\boldsymbol{B}}, \overline{\boldsymbol{C}}) \\
\hat{\boldsymbol{K}}(z) &= \hat{\mathcal{K}}_L(z; \overline{\boldsymbol{A}}, \overline{\boldsymbol{B}}, \overline{\boldsymbol{C}}).
\end{aligned}
$$

Note that

$$\hat{K}_j = \sum_{k=0}^{L-1} \overline{K}_k \exp\left(-2\pi i \frac{jk}{L}\right).$$

Note that this is exactly the same as the Discrete Fourier Transform (DFT):

$$\hat{K} = \mathcal{F}_L K.$$

Therefore $K$ can be recovered from $\hat{K}$ with a single inverse DFT, which requires $O(L \log L)$ operations with the Fast Fourier Transform (FFT) algorithm. □

**Reduction 2: Woodbury Correction**    The primary motivation of Definition 2 is that it turns *powers* of $\overline{A}$ into a single *inverse* of $\overline{A}$ (equation (10)). While DPLR matrices cannot be powered efficiently due to the low-rank term, they can be inverted efficiently by the well-known Woodbury identity.

**Proposition 4** (Binomial Inverse Theorem or Woodbury matrix identity Woodbury (1950); Golub & Van Loan (2013)). *Over a commutative ring $\mathcal{R}$, let $A \in \mathcal{R}^{N \times N}$ and $U, V \in \mathcal{R}^{N \times p}$. Suppose $A$ and $A + UV^*$ are invertible. Then $I_p + V^* A^{-1} U \in \mathcal{R}^{p \times p}$ is invertible and*

$$(A + UV^*)^{-1} = A^{-1} - A^{-1} U (I_p + V^* A^{-1} U)^{-1} V^* A^{-1}$$

With this identity, we can convert the SSM generating function on a DPLR matrix $A$ into one on just its diagonal component.

**Lemma C.3.** *Let $A = \Lambda - PQ^*$ be a diagonal plus low-rank representation. Then for any root of unity $z \in \Omega$, the truncated generating function satisfies*

$$\hat{K}(z) = \frac{2}{1+z}\left[\tilde{C}^* R(z) B - \tilde{C}^* R(z) P \left(1 + Q^* R(z) P\right)^{-1} Q^* R(z) B\right]$$

$$\tilde{C} = C(I - \overline{A}^L)$$

$$R(z; \Lambda) = \left(\frac{2}{\Delta}\frac{1-z}{1+z} - \Lambda\right)^{-1}.$$

*Proof.*  Directly expanding Definition 2 yields

$$\mathcal{K}_L(z; \overline{A}, \overline{B}, \overline{C}) = \overline{C}^* \overline{B} + \overline{C}^* \overline{A}\, \overline{B} z + \cdots + \overline{C}^* \overline{A}^{L-1} \overline{B} z^{L-1}$$

$$= \overline{C}^* \left(I - \overline{A}^L\right)\left(I - \overline{A} z\right)^{-1} \overline{B}$$

$$= \tilde{C}^* \left(I - \overline{A} z\right)^{-1} \overline{B}$$

where $\tilde{C}^* = C^* \left(I - \overline{A}^L\right)$.

We can now explicitly expand the discretized SSM matrices $\overline{A}$ and $\overline{B}$ back in terms of the original SSM parameters $A$ and $B$. Lemma C.4 provides an explicit formula, which allows further simplifying

$$\tilde{C}^* \left(I - \overline{A} z\right)^{-1} \overline{B} = \frac{2}{1+z}\tilde{C}^* \left(\frac{2}{\Delta}\frac{1-z}{1+z} - A\right)^{-1} B$$

$$= \frac{2}{1+z}\tilde{C}^* \left(\frac{2}{\Delta}\frac{1-z}{1+z} - \Lambda + PQ^*\right)^{-1} B$$

$$= \frac{2}{1+z}\left[\tilde{C}^* R(z) B - \tilde{C}^* R(z) P \left(1 + Q^* R(z) P\right)^{-1} Q^* R(z) B\right].$$

The last line applies the Woodbury Identity (Proposition 4) where $R(z) = \left(\frac{2}{\Delta}\frac{1-z}{1+z} - \Lambda\right)^{-1}$.    □

The previous proof used the following self-contained result to back out the original SSM matrices from the discretization.

**Lemma C.4.** *Let* $\overline{A}, \overline{B}$ *be the SSM matrices* $A, B$ *discretized by the bilinear discretization with step size* $\Delta$. *Then*

$$C^* \left(I - \overline{A}z\right)^{-1} \overline{B} = \frac{2\Delta}{1+z} C^* \left[2\frac{1-z}{1+z} - \Delta A\right]^{-1} B$$

*Proof.* Recall that the bilinear discretization that we use (equation (3)) is

$$\overline{A} = \left(I - \frac{\Delta}{2}A\right)^{-1} \left(I + \frac{\Delta}{2}A\right)$$

$$\overline{B} = \left(I - \frac{\Delta}{2}A\right)^{-1} \Delta B$$

The result is proved algebraic manipulations.

$$C^* \left(I - \overline{A}z\right)^{-1} \overline{B} = C^* \left[\left(I - \frac{\Delta}{2}A\right)^{-1}\left(I - \frac{\Delta}{2}A\right) - \left(I - \frac{\Delta}{2}A\right)^{-1}\left(I + \frac{\Delta}{2}A\right)z\right]^{-1} \overline{B}$$

$$= C^* \left[\left(I - \frac{\Delta}{2}A\right) - \left(I + \frac{\Delta}{2}A\right)z\right]^{-1}\left(I - \frac{\Delta}{2}A\right)\overline{B}$$

$$= C^* \left[I(1 - z) - \frac{\Delta}{2}A(1 + z)\right]^{-1} \Delta B$$

$$= \frac{\Delta}{1-z}C^* \left[I - \frac{\Delta A}{2\frac{1-z}{1+z}}\right]^{-1} B$$

$$= \frac{2\Delta}{1+z}C^* \left[2\frac{1-z}{1+z}I - \Delta A\right]^{-1} B$$

$\square$

Note that in the S4 parameterization, instead of constantly computing $\tilde{C} = C\left(I - \overline{A}^L\right)$, we can simply reparameterize our parameters to learn $\tilde{C}$ directly instead of $C$, saving a minor computation cost and simplifying the algorithm.

**Reduction 3: Cauchy Kernel**   We have reduced the original problem of computing $\overline{K}$ to the problem of computing the SSM generating function $\hat{\mathcal{K}}_L(\Omega; \overline{A}, \overline{B}, \overline{C})$ in the case that $\overline{A}$ is a diagonal matrix. We show that this is exactly the same as a Cauchy kernel, which is a well-studied problem with fast and stable numerical algorithms.

**Definition 3.** *A **Cauchy matrix** or kernel on nodes* $\Omega = (\omega_i) \in \mathbb{C}^M$ *and* $\Lambda = (\lambda_j) \in \mathbb{C}^N$ *is*

$$M \in \mathbb{C}^{M \times N} = M(\Omega, \Lambda) = (M_{ij})_{i \in [M], j \in [N]} \qquad M_{ij} = \frac{1}{\omega_i - \lambda_j}.$$

*The computation time of a Cauchy matrix-vector product of size* $M \times N$ *is denoted by* $\mathcal{C}(M, N)$.

Computing with Cauchy matrices is an extremely well-studied problem in numerical analysis, with both fast arithmetic algorithms and fast numerical algorithms based on the famous Fast Multipole Method (FMM) (Pan, 2001; 2015; 2017).

**Proposition 5** (Cauchy). *A Cauchy kernel requires* $O(M + N)$ *space, and operation count*

$$\mathcal{C}(M, N) = \begin{cases} O\left(MN\right) & \text{naively} \\ O\left((M+N)\log^2(M+N)\right) & \text{in exact arithmetic} \\ O\left((M+N)\log(M+N)\log\frac{1}{\varepsilon}\right) & \text{numerically to precision } \varepsilon. \end{cases}$$

**Corollary C.5.** *Evaluating* $Q^* R(\Omega; \Lambda)P$ *(defined in Lemma C.3) for any set of nodes* $\Omega \in \mathbb{C}^L$, *diagonal matrix* $\Lambda$, *and vectors* $P, Q$ *can be computed in* $\mathcal{C}(L, N)$ *operations and* $O(L + N)$ *space, where* $\mathcal{C}(L, N) = \tilde{O}(L + N)$ *is the cost of a Cauchy matrix-vector multiplication.*

*Proof.* For any fixed $\omega \in \Omega$, we want to compute $\sum_j \frac{q_j^* p_j}{\omega - \lambda_j}$. Computing this over all $\omega_i$ is therefore exactly a Cauchy matrix-vector multiplication. □

This completes the proof of Theorem 3. In Algorithm 1, note that the work is dominated by Step 2, which has a constant number of calls to a black-box Cauchy kernel, with complexity given by Proposition 5.

## D  EXPERIMENT DETAILS AND FULL RESULTS

This section contains full experimental procedures and extended results and citations for our experimental evaluation in Section 4. Appendix D.1 corresponds to benchmarking results in Section 4.1, Appendix D.2 corresponds to LRD experiments (LRA and Speech Commands) in Section 4.2, and Appendix D.3 corresponds to the general sequence modeling experiments (generation, image classification, forecasting) in Section 4.3.

### D.1  BENCHMARKING

Benchmarking results from Table 1 and Table 2 were tested on a single A100 GPU.

**Benchmarks against LSSL**    For a given dimension $H$, a single LSSL or S4 layer was constructed with $H$ hidden features. For LSSL, the state size $N$ was set to $H$ as done in (Gu et al., 2021). For S4, the state size $N$ was set to parameter-match the LSSL, which was a state size of $\frac{N}{4}$ due to differences in the parameterization. Table 1 benchmarks a single forward+backward pass of a single layer.

**Benchmarks against Efficient Transformers**    Following (Tay et al., 2021), the Transformer models had 4 layers, hidden dimension 256 with 4 heads, query/key/value projection dimension 128, and batch size 32, for a total of roughly $600k$ parameters. The S4 model was parameter tied while keeping the depth and hidden dimension constant (leading to a state size of $N = 256$).

We note that the relative orderings of these methods can vary depending on the exact hyperparameter settings.

### D.2  LONG-RANGE DEPENDENCIES

This section includes information for reproducing our experiments on the Long-Range Arena and Speech Commands long-range dependency tasks.

**Long Range Arena**    Table 8 contains extended results table with all 11 methods considered in (Tay et al., 2021).

For the S4 model, hyperparameters for all datasets are reported in Table 9. For all datasets, we used the AdamW optimizer with a constant learning rate schedule with decay on validation plateau. However, the learning rate on HiPPO parameters (in particular $\mathbf{\Lambda}, \mathbf{P}, \mathbf{Q}, \mathbf{B}, \mathbf{C}, \Delta$) were reduced to a maximum starting LR of $0.001$, which improves stability since the HiPPO equation is crucial to performance.

The S4 state size was always fixed to $N = 64$.

As S4 is a sequence-to-sequence model with output shape (batch, length, dimension) and LRA tasks are classification, mean pooling along the length dimension was applied after the last layer.

We note that most of these results were trained for far longer than what was necessary to achieve SotA results (e.g., the `Image` task reaches SotA in 1 epoch). Results often keep improving with longer training times.

**Hardware.** All models were run on single GPU. Some tasks used an A100 GPU (notably, the Path-X experiments), which has a larger max memory of 40Gb. To reproduce these on smaller GPUs, the batch size can be reduced or gradients can be accumulated for two batches.

**Path-X.** We remark that an earlier version of this paper reported a higher score for Path-X. This earlier version used a different variant of the dataset, due to a misunderstanding of the properties of the dataset. More specifically, we found that S4 scored $93.68\%$ accuracy on a version that involved

Table 8: Full results for the Long Range Arena (LRA) benchmark for long-range dependencies in sequence models. (Top): Original Transformer variants in LRA. (Bottom): Other models reported in the literature.

| Model | LISTOPS | TEXT | RETRIEVAL | IMAGE | PATHFINDER | PATH-X | AVG |
|---|---|---|---|---|---|---|---|
| Random | 10.00 | 50.00 | 50.00 | 10.00 | 50.00 | 50.00 | 36.67 |
| Transformer | 36.37 | 64.27 | 57.46 | 42.44 | 71.40 | ✗ | 53.66 |
| Local Attention | 15.82 | 52.98 | 53.39 | 41.46 | 66.63 | ✗ | 46.71 |
| Sparse Trans. | 17.07 | 63.58 | 59.59 | 44.24 | 71.71 | ✗ | 51.03 |
| Longformer | 35.63 | 62.85 | 56.89 | 42.22 | 69.71 | ✗ | 52.88 |
| Linformer | 35.70 | 53.94 | 52.27 | 38.56 | 76.34 | ✗ | 51.14 |
| Reformer | 37.27 | 56.10 | 53.40 | 38.07 | 68.50 | ✗ | 50.56 |
| Sinkhorn Trans. | 33.67 | 61.20 | 53.83 | 41.23 | 67.45 | ✗ | 51.23 |
| Synthesizer | 36.99 | 61.68 | 54.67 | 41.61 | 69.45 | ✗ | 52.40 |
| BigBird | 36.05 | 64.02 | 59.29 | 40.83 | 74.87 | ✗ | 54.17 |
| Linear Trans. | 16.13 | 65.90 | 53.09 | 42.34 | 75.30 | ✗ | 50.46 |
| Performer | 18.01 | 65.40 | 53.82 | 42.77 | 77.05 | ✗ | 51.18 |
| FNet | 35.33 | 65.11 | 59.61 | 38.67 | 77.80 | ✗ | 54.42 |
| Nyströmformer | 37.15 | 65.52 | 79.56 | 41.58 | 70.94 | ✗ | 57.46 |
| Luna-256 | 37.25 | 64.57 | 79.29 | 47.38 | 77.72 | ✗ | 59.37 |
| **S4** | **58.35** | **76.02** | **87.09** | **87.26** | **86.05** | **88.10** | **80.48** |

Table 9: The values of the best hyperparameters found for classification datasets; LRA (Top) and images/speech (Bottom). LR is learning rate and WD is weight decay. BN and LN refer to Batch Normalization and Layer Normalization.

| | Depth | Features $H$ | Norm | Pre-norm | Dropout | LR | Batch Size | Epochs | WD | Patience |
|---|---|---|---|---|---|---|---|---|---|---|
| **ListOps** | 6 | 128 | BN | False | 0 | 0.01 | 100 | 50 | 0.01 | 5 |
| **Text** | 4 | 64 | BN | True | 0 | 0.001 | 50 | 20 | 0 | 5 |
| **Retrieval** | 6 | 256 | BN | True | 0 | 0.002 | 64 | 20 | 0 | 20 |
| **Image** | 6 | 512 | LN | False | 0.2 | 0.004 | 50 | 200 | 0.01 | 20 |
| **Pathfinder** | 6 | 256 | BN | True | 0.1 | 0.004 | 100 | 200 | 0 | 10 |
| **Path-X** | 6 | 256 | BN | True | 0.0 | 0.0005 | 32 | 100 | 0 | 20 |
| **CIFAR-10** | 6 | 1024 | LN | False | 0.25 | 0.01 | 50 | 200 | 0.01 | 20 |
| **Speech Commands (MFCC)** | 4 | 256 | LN | False | 0.2 | 0.01 | 100 | 50 | 0 | 5 |
| **Speech Commands (Raw)** | 6 | 128 | BN | True | 0.1 | 0.01 | 20 | 150 | 0 | 10 |

taking the $256 \times 256$ resolution version of the Pathfinder dataset and averaging every $2 \times 2$ square; we erroneously thought that this version of the dataset was equivalent to the original Path-X.

After discussions with the LRA authors, we discovered that this is not equivalent to the $128 \times 128$ resolution Pathfinder dataset (the correct Path-X), which is in fact much harder. In fact, Path-X is so difficult that a 2D CNN without global receptive field (e.g. ResNet-18 or ResNet-34) also cannot achieve above chance. This fact led to the original misunderstanding, as we could not solve this image classification task even with a ResNet and thought the data might have errors.

**Speech Commands** We provide details of sweeps run for baseline methods run by us—numbers for all others method are taken from Gu et al. (2021). The best hyperparameters used for S4 are included in Table 9.

*Transformer (Vaswani et al., 2017)* For MFCC, we swept the number of model layers $\{2, 4\}$, dropout $\{0, 0.1\}$ and learning rates $\{0.001, 0.0005\}$. We used 8 attention heads, model dimension 128, prenorm, positional encodings, and trained for 150 epochs with a batch size of 100. For Raw, the Transformer model's memory usage made training impossible.

*Performer (Choromanski et al., 2020)* For MFCC, we swept the number of model layers $\{2, 4\}$, dropout $\{0, 0.1\}$ and learning rates $\{0.001, 0.0005\}$. We used 8 attention heads, model dimension 128, prenorm, positional encodings, and trained for 150 epochs with a batch size of 100. For Raw, we used a model dimension of 128, 4 attention heads, prenorm, and a batch size of 16. We reduced the number of model layers to 4, so the model would fit on the single GPU. We trained for 100 epochs with a learning rate of 0.001 and no dropout.

*ExpRNN (Lezcano-Casado & Martínez-Rubio, 2019)* For MFCC, we swept hidden sizes $\{256, 512\}$ and learning rates $\{0.001, 0.002, 0.0005\}$. Training was run for 200 epochs, with a single layer model using a batch size of 100. For Raw, we swept hidden sizes $\{32, 64\}$ and learning rates $\{0.001, 0.0005\}$ (however, ExpRNN failed to learn).

*Lipschitz RNN (Erichson et al., 2021)* For MFCC, we swept hidden sizes $\{256, 512\}$ and learning rates $\{0.001, 0.002, 0.0005\}$. Training was run for 150 epochs, with a single layer model using a batch size of 100. For Raw, we found that LipschitzRNN was too slow to train on a single GPU (requiring a full day for 1 epoch of training alone).

*WaveGAN Discriminator (Donahue et al., 2019)* The WaveGAN-D in Table 4 is actually our improved version of the discriminator network from the recent WaveGAN model for speech (Donahue et al., 2019). This CNN actually did not work well out-of-the-box, and we added several features to help it perform better. The final model is highly specialized compared to our model, and includes:

- Downsampling or pooling between layers, induced by strided convolutions, that decrease the sequence length between layers.
- A global fully-connected output layer; thus the model only works for one input sequence length and does not work on MFCC features or the frequency-shift setting in Table 4.
- Batch Normalization is essential, whereas S4 works equally well with either Batch Normalization or Layer Normalization.
- Almost $90\times$ as many parameters as the S4 model (26.3M vs. 0.3M).

### D.3 GENERAL SEQUENCE MODELING

This subsection corresponds to the experiments in Section 4.3. Because of the number of experiments in this section, we use subsubsection dividers for different tasks to make it easier to follow: CIFAR-10 density estimation Appendix D.3.1, WikiText-103 language modeling Appendix D.3.2, autoregressive generation Appendix D.3.3, sequential image classification Appendix D.3.4, and time-series forecasting Appendix D.3.5.

#### D.3.1 CIFAR DENSITY ESTIMATION

This task used a different backbone than the rest of our experiments. We used blocks of alternating S4 layers and position-wise feed-forward layers (in the style of Transformer blocks). Each feed-forward intermediate dimension was set to $2\times$ the hidden size of the incoming S4 layer. Similar to Salimans et al. (2017), we used a UNet-style backbone consisting of $B$ identical blocks followed by a downsampling layer. The downsampling rates were $3, 4, 4$ (the 3 chosen because the sequence consists of RGB pixels). The base model had $B = 8$ with starting hidden dimension 128, while the large model had $B = 16$ with starting hidden dimension 192.

We experimented with both the mixture of logistics from (Salimans et al., 2017) as well as a simpler 256-way categorical loss. We found they were pretty close and ended up using the simpler softmax loss along with using input embeddings.

We used the LAMB optimizer with learning rate 0.005. The base model had no dropout, while the large model had dropout 0.1 before the linear layers inside the S4 and FF blocks.

#### D.3.2 WIKITEXT-103 LANGUAGE MODELING

The RNN baselines included in Table 7 are the AWD-QRNN (Merity et al., 2018), an efficient linear gated RNN, and the LSTM + Cache + Hebbian + MbPA (Rae et al., 2018), the best performing pure RNN in the literature. The CNN baselines are the CNN with GLU activations (Dauphin et al., 2017), the TrellisNet (Bai et al., 2019), Dynamic Convolutions (Wu et al., 2019), and TaLK Convolutions (Lioutas & Guo, 2020).

The Transformer baseline is (Baevski & Auli, 2018), which uses Adaptive Inputs with a tied Adaptive Softmax. This model is a standard high-performing Transformer baseline on this benchmark, used for example by Lioutas & Guo (2020) and many more.

Our S4 model uses the same Transformer backbone as in (Baevski & Auli, 2018). The model consists of 16 blocks of S4 layers alternated with position-wise feedforward layers, with a feature dimension of 1024. Because our S4 layer has around 1/4 the number of parameters as a self-attention layer

with the same dimension, we made two modifications to match the parameter count better: (i) we used a GLU activation after the S4 linear layer (Section 3.4) (ii) we used two S4 layers per block. Blocks use Layer Normalization in the pre-norm position. The embedding and softmax layers were the Adaptive Embedding from (Baevski & Auli, 2018) with standard cutoffs 20000, 40000, 200000.

Evaluation was performed similarly to the basic setting in (Baevski & Auli, 2018), Table 5, which involves sliding non-overlapping windows of width 1024 tokens. Other settings are reported in (Baevski & Auli, 2018) that include more context at training and evaluation time and improves the score. Because such evaluation protocols are orthogonal to the basic model, we do not consider them and report the base score from (Baevski & Auli, 2018) Table 5.

Instead of SGD+Momentum with multiple cosine learning rate annealing cycles, our S4 model was trained with the simpler AdamW optimizer with a single cosine learning rate cycle with a maximum of 800000 steps. The initial learning rate was set to 0.0005. We used 8 A100 GPUs with a batch size of 8 per gpu and context size 1024. We used no gradient clipping and a weight decay of 0.1. Unlike (Baevski & Auli, 2018) which specified different dropout rates for different parameters, we used a constant dropout rate of 0.25 throughout the network, including before every linear layer and on the residual branches.

### D.3.3 Autoregressive Generation Speed

**Protocol.** To account for different model sizes and memory requirements for each method, we benchmark generation speed by throughput, measured in images per second (Table 6) or tokens per second (Table 7). Each model generates images on a single $A100$ GPU, maximizing batch size to fit in memory. (For CIFAR-10 generation we limited memory to 16Gb, to be more comparable to the Transformer and Linear Transformer results reported from (Katharopoulos et al., 2020).)

**Baselines.** The Transformer and Linear Transformer baselines reported in Table 6 are the results reported directly from Katharopoulos et al. (2020). Note that the Transformer number is the one in their Appendix, which implements the optimized cached implementation of self-attention.

For all other baseline models, we used open source implementations of the models to benchmark generation speed. For the PixelCNN++, we used the fast cached version by Ramachandran et al. (2017), which sped up generation by orders of magnitude from the naive implementation. This code was only available in TensorFlow, which may have slight differences compared to the rest of the baselines which were implemented in PyTorch.

We were unable to run the Sparse Transformer (Child et al., 2019) model due to issues with their custom CUDA implementation of the sparse attention kernel, which we were unable to resolve.

The Transformer baseline from Table 7 was run using a modified GPT-2 backbone from the HuggingFace repository, configured to recreate the architecture reported in (Baevski & Auli, 2018). These numbers are actually slightly favorable to the baseline, as we did not include the timing of the embedding or softmax layers, whereas the number reported for S4 is the full model.

### D.3.4 Pixel-Level Sequential Image Classification

Our models were trained with the AdamW optimizer for up to 200 epochs. Hyperparameters for the CIFAR-10 model is reported in Table 9.

For our comparisons against ResNet-18, the main differences between the base models are that S4 uses LayerNorm by default while ResNet uses BatchNorm. The last ablation in Section 4.3 swaps the normalization type, using BatchNorm for S4 and LayerNorm for ResNet, to ablate this architectural difference. The experiments with augmentation take the base model and train with mild data augmentation: horizontal flips and random crops (with symmetric padding).

### D.3.5 Time Series Forecasting compared to Informer

We include a simple figure (Fig. 3) contrasting the architecture of S4 against that of the Informer (Zhou et al., 2021).

In Fig. 3, the goal is to forecast a contiguous range of future predictions (Green, length $F$) given a range of past context (Blue, length $C$). We simply concatenate the entire context with a sequence of masks set to the length of the forecast window. This input is a single sequence of length $C + F$ that

Table 10: (**Pixel-level image classification.**) Citations refer to the original model; additional citation indicates work from which this baseline is reported.

| Model | sMNIST | pMNIST | sCIFAR |
|---|---|---|---|
| Transformer (Vaswani et al., 2017; Trinh et al., 2018) | 98.9 | 97.9 | 62.2 |
| CKConv (Romero et al., 2021) | 99.32 | 98.54 | 63.74 |
| TrellisNet (Bai et al., 2019) | 99.20 | 98.13 | 73.42 |
| TCN (Bai et al., 2018) | 99.0 | 97.2 | - |
| LSTM (Hochreiter & Schmidhuber, 1997; Gu et al., 2020b) | 98.9 | 95.11 | 63.01 |
| r-LSTM  (Trinh et al., 2018) | 98.4 | 95.2 | 72.2 |
| Dilated GRU (Chang et al., 2017) | 99.0 | 94.6 | - |
| Dilated RNN (Chang et al., 2017) | 98.0 | 96.1 | - |
| IndRNN (Li et al., 2018) | 99.0 | 96.0 | - |
| expRNN (Lezcano-Casado & Martínez-Rubio, 2019) | 98.7 | 96.6 | - |
| UR-LSTM | 99.28 | 96.96 | 71.00 |
| UR-GRU (Gu et al., 2020b) | 99.27 | 96.51 | 74.4 |
| LMU (Voelker et al., 2019) | - | 97.15 | - |
| HiPPO-RNN (Gu et al., 2020a) | 98.9 | 98.3 | 61.1 |
| UNIcoRNN (Rusch & Mishra, 2021) | - | 98.4 | - |
| LMUFFT (Chilkuri & Eliasmith, 2021) | - | 98.49 | - |
| LipschitzRNN (Erichson et al., 2021) | 99.4 | 96.3 | 64.2 |
| **S4** | **99.63** | **98.70** | **91.13** |

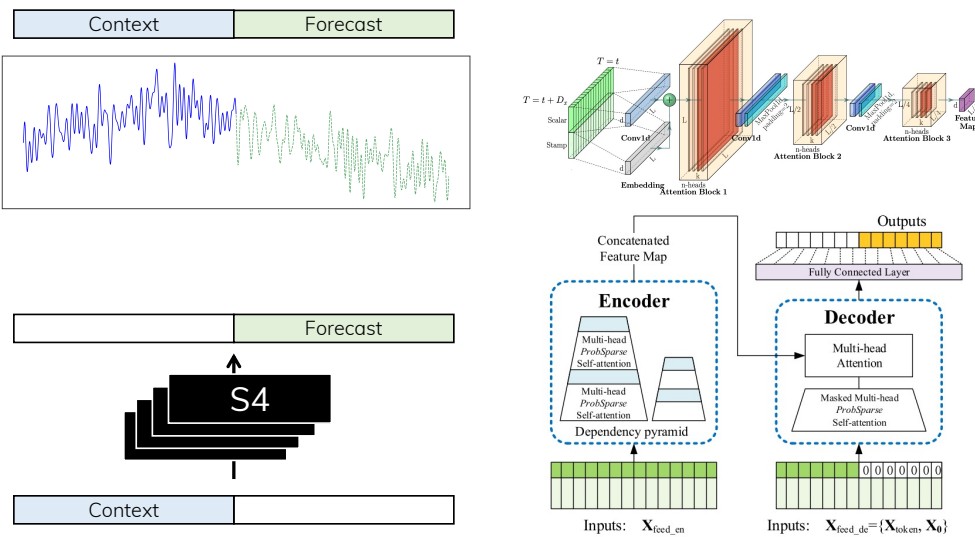

Figure 3: Comparison of S4 and specialized time-series models for forecasting tasks. (*Top Left*) The forecasting task involves predicting future values of a time-series given past context. (*Bottom Left*) We perform simple forecasting using a sequence model such S4 as a black box. (*Right*) Informer uses an encoder-decoder architecture designed specifically for forecasting problems involving a customized attention module (figure taken from Zhou et al. (2021)).

is run through the same simple deep S4 model used throughout this work, which maps to an output of length $C + F$. We then use just the last $F$ features as the forecasted predictions.

Tables 11 and 12 contain full results on all 50 settings considered by Zhou et al. (2021). S4 sets the best results on 40 out of 50 of these settings.

| Methods | S4 | | Informer | | Informer† | | LogTrans | | Reformer | | LSTMa | | DeepAR | | ARIMA | | Prophet | |
|---|---|---|---|---|---|---|---|---|---|---|---|---|---|---|---|---|---|---|
| Metric | MSE | MAE | MSE | MAE | MSE | MAE | MSE | MAE | MSE | MAE | MSE | MAE | MSE | MAE | MSE | MAE | MSE | MAE |
| ETTh₁ 24 | **0.061** | **0.191** | 0.098 | 0.247 | 0.092 | 0.246 | 0.103 | 0.259 | 0.222 | 0.389 | 0.114 | 0.272 | 0.107 | 0.280 | 0.108 | 0.284 | 0.115 | 0.275 |
| ETTh₁ 48 | **0.079** | **0.220** | 0.158 | 0.319 | 0.161 | 0.322 | 0.167 | 0.328 | 0.284 | 0.445 | 0.193 | 0.358 | 0.162 | 0.327 | 0.175 | 0.424 | 0.168 | 0.330 |
| ETTh₁ 168 | **0.104** | **0.258** | 0.183 | 0.346 | 0.187 | 0.355 | 0.207 | 0.375 | 1.522 | 1.191 | 0.236 | 0.392 | 0.239 | 0.422 | 0.396 | 0.504 | 1.224 | 0.763 |
| ETTh₁ 336 | **0.080** | **0.229** | 0.222 | 0.387 | 0.215 | 0.369 | 0.230 | 0.398 | 1.860 | 1.124 | 0.590 | 0.698 | 0.445 | 0.552 | 0.468 | 0.593 | 1.549 | 1.820 |
| ETTh₁ 720 | **0.116** | **0.271** | 0.269 | 0.435 | 0.257 | 0.421 | 0.273 | 0.463 | 2.112 | 1.436 | 0.683 | 0.768 | 0.658 | 0.707 | 0.659 | 0.766 | 2.735 | 3.253 |
| ETTh₂ 24 | 0.095 | 0.234 | **0.093** | **0.240** | 0.099 | 0.241 | 0.102 | 0.255 | 0.263 | 0.437 | 0.155 | 0.307 | 0.098 | 0.263 | 3.554 | 0.445 | 0.199 | 0.381 |
| ETTh₂ 48 | 0.191 | 0.346 | **0.155** | **0.314** | 0.159 | 0.317 | 0.169 | 0.348 | 0.458 | 0.545 | 0.190 | 0.348 | 0.163 | 0.341 | 3.190 | 0.474 | 0.304 | 0.462 |
| ETTh₂ 168 | **0.167** | **0.333** | 0.232 | 0.389 | 0.235 | 0.390 | 0.246 | 0.422 | 1.029 | 0.879 | 0.385 | 0.514 | 0.255 | 0.414 | 2.800 | 0.595 | 2.145 | 1.068 |
| ETTh₂ 336 | **0.189** | **0.361** | 0.263 | 0.417 | 0.258 | 0.423 | 0.267 | 0.437 | 1.668 | 1.228 | 0.558 | 0.606 | 0.604 | 0.607 | 2.753 | 0.738 | 2.096 | 2.543 |
| ETTh₂ 720 | **0.187** | **0.358** | 0.277 | 0.431 | 0.285 | 0.442 | 0.303 | 0.493 | 2.030 | 1.721 | 0.640 | 0.681 | 0.429 | 0.580 | 2.878 | 1.044 | 3.355 | 4.664 |
| ETTm₁ 24 | **0.024** | **0.117** | 0.030 | 0.137 | 0.034 | 0.160 | 0.065 | 0.202 | 0.095 | 0.228 | 0.121 | 0.233 | 0.091 | 0.243 | 0.090 | 0.206 | 0.120 | 0.290 |
| ETTm₁ 48 | **0.051** | **0.174** | 0.069 | 0.203 | 0.066 | 0.194 | 0.078 | 0.220 | 0.249 | 0.390 | 0.305 | 0.411 | 0.219 | 0.362 | 0.179 | 0.306 | 0.133 | 0.305 |
| ETTm₁ 96 | **0.086** | **0.229** | 0.194 | 0.372 | 0.187 | 0.384 | 0.199 | 0.386 | 0.920 | 0.767 | 0.287 | 0.420 | 0.364 | 0.496 | 0.272 | 0.399 | 0.194 | 0.396 |
| ETTm₁ 288 | **0.160** | **0.327** | 0.401 | 0.554 | 0.409 | 0.548 | 0.411 | 0.572 | 1.108 | 1.245 | 0.524 | 0.584 | 0.948 | 0.795 | 0.462 | 0.558 | 0.452 | 0.574 |
| ETTm₁ 672 | **0.292** | **0.466** | 0.512 | 0.644 | 0.519 | 0.665 | 0.598 | 0.702 | 1.793 | 1.528 | 1.064 | 0.873 | 2.437 | 1.352 | 0.639 | 0.697 | 2.747 | 1.174 |
| Weather 24 | 0.125 | 0.254 | **0.117** | 0.251 | 0.119 | 0.256 | 0.136 | 0.279 | 0.231 | 0.401 | 0.131 | 0.254 | 0.128 | 0.274 | 0.219 | 0.355 | 0.302 | 0.433 |
| Weather 48 | 0.181 | **0.305** | **0.178** | 0.318 | 0.185 | 0.316 | 0.206 | 0.356 | 0.328 | 0.423 | 0.190 | 0.334 | 0.203 | 0.353 | 0.273 | 0.409 | 0.445 | 0.536 |
| Weather 168 | **0.198** | **0.333** | 0.266 | 0.398 | 0.269 | 0.404 | 0.309 | 0.439 | 0.654 | 0.634 | 0.341 | 0.448 | 0.293 | 0.451 | 0.503 | 0.599 | 2.441 | 1.142 |
| Weather 336 | 0.300 | 0.417 | **0.297** | **0.416** | 0.310 | 0.422 | 0.359 | 0.484 | 1.792 | 1.093 | 0.456 | 0.554 | 0.585 | 0.644 | 0.728 | 0.730 | 1.987 | 2.468 |
| Weather 720 | **0.245** | **0.375** | 0.359 | 0.466 | 0.361 | 0.471 | 0.388 | 0.499 | 2.087 | 1.534 | 0.866 | 0.809 | 0.499 | 0.596 | 1.062 | 0.943 | 3.859 | 1.144 |
| ECL 48 | 0.222 | **0.350** | 0.239 | 0.359 | 0.238 | 0.368 | 0.280 | 0.429 | 0.971 | 0.884 | 0.493 | 0.539 | **0.204** | 0.357 | 0.879 | 0.764 | 0.524 | 0.595 |
| ECL 168 | 0.331 | **0.421** | 0.447 | 0.503 | 0.442 | 0.514 | 0.454 | 0.529 | 1.671 | 1.587 | 0.723 | 0.655 | **0.315** | 0.436 | 1.032 | 0.833 | 2.725 | 1.273 |
| ECL 336 | **0.328** | **0.422** | 0.489 | 0.528 | 0.501 | 0.552 | 0.514 | 0.563 | 3.528 | 2.196 | 1.212 | 0.898 | 0.414 | 0.519 | 1.136 | 0.876 | 2.246 | 3.077 |
| ECL 720 | **0.428** | **0.494** | 0.540 | 0.571 | 0.543 | 0.578 | 0.558 | 0.609 | 4.891 | 4.047 | 1.511 | 0.966 | 0.563 | 0.595 | 1.251 | 0.933 | 4.243 | 1.415 |
| ECL 960 | **0.432** | **0.497** | 0.582 | 0.608 | 0.594 | 0.638 | 0.624 | 0.645 | 7.019 | 5.105 | 1.545 | 1.006 | 0.657 | 0.683 | 1.370 | 0.982 | 6.901 | 4.264 |
| Count | 22 | | 5 | | 0 | | 0 | | 0 | | 0 | | 2 | | 0 | | 0 | |

Table 11: Univariate long sequence time-series forecasting results on four datasets (five cases).

| Methods | S4 | | Informer | | Informer† | | LogTrans | | Reformer | | LSTMa | | LSTnet | |
|---|---|---|---|---|---|---|---|---|---|---|---|---|---|---|
| Metric | MSE | MAE | MSE | MAE | MSE | MAE | MSE | MAE | MSE | MAE | MSE | MAE | MSE | MAE |
| ETTh₁ 24 | **0.525** | **0.542** | 0.577 | 0.549 | 0.620 | 0.577 | 0.686 | 0.604 | 0.991 | 0.754 | 0.650 | 0.624 | 1.293 | 0.901 |
| ETTh₁ 48 | **0.641** | **0.615** | 0.685 | 0.625 | 0.692 | 0.671 | 0.766 | 0.757 | 1.313 | 0.906 | 0.702 | 0.675 | 1.456 | 0.960 |
| ETTh₁ 168 | 0.980 | 0.779 | **0.931** | **0.752** | 0.947 | 0.797 | 1.002 | 0.846 | 1.824 | 1.138 | 1.212 | 0.867 | 1.997 | 1.214 |
| ETTh₁ 336 | 1.407 | 0.910 | 1.128 | 0.873 | **1.094** | **0.813** | 1.362 | 0.952 | 2.117 | 1.280 | 1.424 | 0.994 | 2.655 | 1.369 |
| ETTh₁ 720 | **1.162** | **0.842** | 1.215 | 0.896 | 1.241 | 0.917 | 1.397 | 1.291 | 2.415 | 1.520 | 1.960 | 1.322 | 2.143 | 1.380 |
| ETTh₂ 24 | 0.871 | 0.736 | **0.720** | **0.665** | 0.753 | 0.727 | 0.828 | 0.750 | 1.531 | 1.613 | 1.143 | 0.813 | 2.742 | 1.457 |
| ETTh₂ 48 | **1.240** | **0.867** | 1.457 | 1.001 | 1.461 | 1.077 | 1.806 | 1.034 | 1.871 | 1.735 | 1.671 | 1.221 | 3.567 | 1.687 |
| ETTh₂ 168 | **2.580** | **1.255** | 3.489 | 1.515 | 3.485 | 1.612 | 4.070 | 1.681 | 4.660 | 1.846 | 4.117 | 1.674 | 3.242 | 2.513 |
| ETTh₂ 336 | **1.980** | **1.128** | 2.723 | 1.340 | 2.626 | 1.285 | 3.875 | 1.763 | 4.028 | 1.688 | 3.434 | 1.549 | 2.544 | 2.591 |
| ETTh₂ 720 | **2.650** | **1.340** | 3.467 | 1.473 | 3.548 | 1.495 | 3.913 | 1.552 | 5.381 | 2.015 | 3.963 | 1.788 | 4.625 | 3.709 |
| ETTm₁ 24 | 0.426 | 0.487 | 0.323 | **0.369** | **0.306** | 0.371 | 0.419 | 0.412 | 0.724 | 0.607 | 0.621 | 0.629 | 1.968 | 1.170 |
| ETTm₁ 48 | 0.580 | 0.565 | 0.494 | 0.503 | **0.465** | **0.470** | 0.507 | 0.583 | 1.098 | 0.777 | 1.392 | 0.939 | 1.999 | 1.215 |
| ETTm₁ 96 | 0.699 | 0.649 | **0.678** | 0.614 | 0.681 | **0.612** | 0.768 | 0.792 | 1.433 | 0.945 | 1.339 | 0.913 | 2.762 | 1.542 |
| ETTm₁ 288 | **0.824** | **0.674** | 1.056 | 0.786 | 1.162 | 0.879 | 1.462 | 1.320 | 1.820 | 1.094 | 1.740 | 1.124 | 1.257 | 2.076 |
| ETTm₁ 672 | **0.846** | **0.709** | 1.192 | 0.926 | 1.231 | 1.103 | 1.669 | 1.461 | 2.187 | 1.232 | 2.736 | 1.555 | 1.917 | 2.941 |
| Weather 24 | **0.334** | 0.385 | 0.335 | **0.381** | 0.349 | 0.397 | 0.435 | 0.477 | 0.655 | 0.583 | 0.546 | 0.570 | 0.615 | 0.545 |
| Weather 48 | 0.406 | 0.444 | 0.395 | 0.459 | **0.386** | **0.433** | 0.426 | 0.495 | 0.729 | 0.666 | 0.829 | 0.677 | 0.660 | 0.589 |
| Weather 168 | **0.525** | **0.527** | 0.608 | 0.567 | 0.613 | 0.582 | 0.727 | 0.671 | 1.318 | 0.855 | 1.038 | 0.835 | 0.748 | 0.647 |
| Weather 336 | **0.531** | **0.539** | 0.702 | 0.620 | 0.707 | 0.634 | 0.754 | 0.670 | 1.930 | 1.167 | 1.657 | 1.059 | 0.782 | 0.683 |
| Weather 720 | **0.578** | **0.578** | 0.831 | 0.731 | 0.834 | 0.741 | 0.885 | 0.773 | 2.726 | 1.575 | 1.536 | 1.109 | 0.851 | 0.757 |
| ECL 48 | **0.255** | **0.352** | 0.344 | 0.393 | 0.334 | 0.399 | 0.355 | 0.418 | 1.404 | 0.999 | 0.486 | 0.572 | 0.369 | 0.445 |
| ECL 168 | **0.283** | **0.373** | 0.368 | 0.424 | 0.353 | 0.420 | 0.368 | 0.432 | 1.515 | 1.069 | 0.574 | 0.602 | 0.394 | 0.476 |
| ECL 336 | **0.292** | **0.382** | 0.381 | 0.431 | 0.381 | 0.439 | 0.373 | 0.439 | 1.601 | 1.104 | 0.886 | 0.795 | 0.419 | 0.477 |
| ECL 720 | **0.289** | **0.377** | 0.406 | 0.443 | 0.391 | 0.438 | 0.409 | 0.454 | 2.009 | 1.170 | 1.676 | 1.095 | 0.556 | 0.565 |
| ECL 960 | **0.299** | **0.387** | 0.460 | 0.548 | 0.492 | 0.550 | 0.477 | 0.589 | 2.141 | 1.387 | 1.591 | 1.128 | 0.605 | 0.599 |
| Count | 18 | | 5 | | 6 | | 0 | | 0 | | 0 | | 0 | |

Table 12: Multivariate long sequence time-series forecasting results on four datasets (five cases).

## D.4 VISUALIZATIONS

We visualize the convolutional filter $\bar{K}$ learned by S4 for the Pathfinder and CIFAR-10 tasks in Appendix D.4.

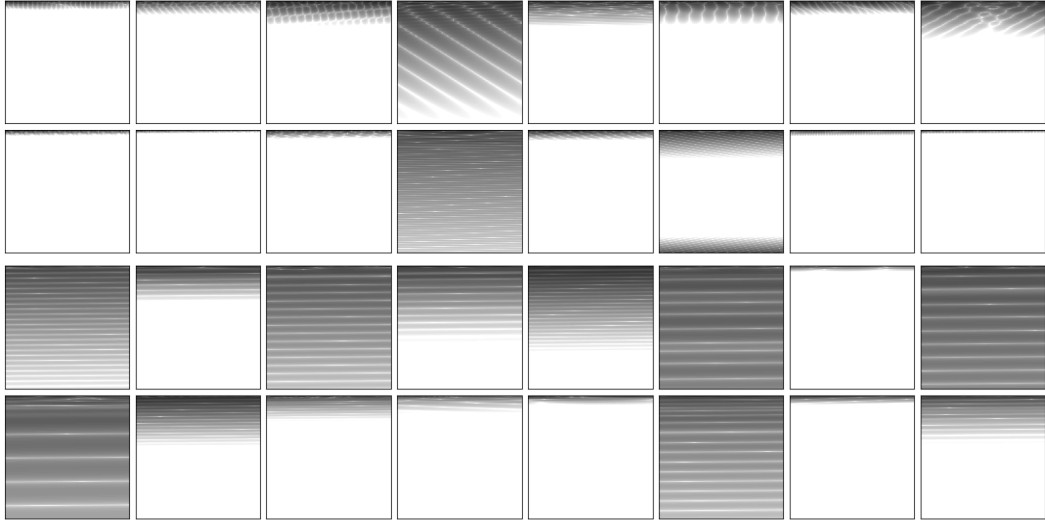

Figure 4: (**Convolutional filters on Pathfinder**) A random selection of filters learned by S4 in the first layer (top 2 rows) and last layer (bottom 2 rows) of the best model.

