# OpenReview forum: "Efficiently Modeling Long Sequences with Structured State Spaces"
_ICLR.cc/2022/Conference — ICLR 2022 Oral_

### Official Review · Reviewer_7JCQ · 2021-10-31

**Correctness:** 4
**Technical Novelty And Significance:** 4
**Empirical Novelty And Significance:** 4
**Recommendation:** 8
**Confidence:** 4

**Main Review:**

Strengths:
* The motivation for the approach is clear both from a high-level as well as mathematical viewpoint.  The authors do a good job walking through the theoretical justification and solution of the approach and proposed algorithm.

* The discussion and analysis is performed on a broad domain of tasks.  As the authors discuss, many approaches today are narrowly focused around a single task/domain which limits our broader understanding of LRD modeling.  By using such a varied set of experiments, the usefulness of this approach is well highlighted.

* The introduction is very well written.  The general reader may be less familiar with some of these past works and the authors do an excellent job highlighting the motivation and current status of efforts in this line of study.  Similarly the background is clear and detailed without being unnecessarily complicated or verbose.

* The performance is compelling from both an efficiency and accuracy standpoint across a number of tasks.

------------------------

Weaknesses:
* Limitations and next steps aren’t explicitly discussed.  It would be helpful for the authors to include even a few sentences on this.

* A known issue with approaches like transformers is the need for huge datasets (even by deep learning standards). How does the performance of this approach vary as a function of dataset size?  Does this method work in a low-data regime?

* The results section is more difficult to follow, especially compared to the rest of the text.  Many of the results are included in the appendix (which is fine especially given their extensiveness).  However, some of the table references are strange.  Table 4 is given, but not directly referenced in the main text (should go with Raw Speech Classification on p.8).  Table 5 is not referenced until after Table 6.  The section "Irregularly Sampled Data" doesn't explicitly reference where those results can be found (i.e. Table 4).  This makes the results section a little bit unclear as the reader has to the text-to-table mapping.   Table 4 caption is a bit unclear (i.e. requires going to the text to interpret what things mean).

* Figure 2 (feature visualization): the authors state that the visualization of the low-level features shows that context over a small area is being learned and the rest of the image is ignored.  However, two of the activations seem to span the entire kernel.  Similarly, one of the higher-level filters is over only a small area.  This seems to be more of a general trend (low learns mostly, but not completely localized, high learns mostly, but not completely global), but not a hard and fast rule.  Is there a hypothesis as to why only some, but not all filters in these layers follow these trends?  With the current visualization, a number of the filters appear identical (just a few solid rows at the top)- are these degenerate or is this a limitation of the visualization?  There is less “hierarchical” structural differences in the convolutional filters (Figure 5 appendix).  Could the authors explain this?

* Minor: Figure 1, I missed the green K several times (spent a while looking for the green in the figure).  Perhaps altering the perceptual qualities of the font could help draw attention to these variables more.


**Summary Of The Paper:**

The authors propose a sequence modeling approach called the structured state space model (S3) which parameterized the SSM in a more computationally efficient manner.  This is done through decomposing the structured state matrices A into a low-rank and skew-symmetric term and expanding the SSM in frequency space and using a multipole-like evaluation.  This approach maintains much of the gains and efficiencies of past SSM approaches while being more computationally stable and efficient as demonstrated on several tasks across broad domains (speech, images, text).  The paper first steps through the theoretical motivations and justifications of this approach and then performs a number of experiments to demonstrate the competitive or superior performance on a wide range of LRD tasks.

**Summary Of The Review:**

My recommendation is to accept this work.  This paper proposes a novel parameterization for solving SSM which provides computational efficiency and accuracy gains.  The approach is explained and justified theoretically and the performance is evaluated on a number of tasks across different domains.  The text is clear and well written.  The authors explicitly state the goal of finding general methods that work on a broad range of tasks and this approach offers a good step toward solving that problem under realistic computational constraints.

---

> ### Author Response · Authors · 2021-11-20
> **Response to Reviewer 7JCQ**
>
> We sincerely appreciate the reviewer’s thorough and thoughtful review. Overall, the reviewer found our paper well-written and the model’s performance compelling. The reviewer’s comments suggested ways to improve the presentation, which we respond to in order:
>
> 1.  Our updated manuscript includes a discussion of limitations and next steps. This includes a remark on where S4 has had the most difficulties so far (text data), as well as some ideas for extensions on both theory and applications.
>
> 2. Some of our tasks had relatively low data, such as the sequential image tasks (50k train examples), some LRA tasks, and the time-series forecasting tasks (where each epoch took only a few seconds) where S4 consistently outperformed the Informer baseline. We hypothesize that S4 performs well in low-data regimes on time-series data because SSMs are a continuous-time model and have strong inductive bias for such time-series data.
>
> 3. We thank the reviewer for numerous presentation suggestions in the experiments. We have improved many of the references between tables and text and the ordering of tables, although some elements of the previous draft were preserved for formatting and space considerations.
>
> 4. Indeed, the trends in the context size in lower versus higher layers are not hard and fast rules. One explanation is that unlike CNNs which use local filters and pooling -- and are very explicitly designed to be hierarchical -- our S4 model is more generic, and each layer is allowed to learn dependencies of any length. Any trends that appear are emergent properties of the model, and investigating the learned representations in more detail is an interesting question for future study.

---

> > ### Comment · Reviewer_7JCQ · 2021-11-29
> > **Thank you for response**
> >
> > Thank you for your answers to the questions raised.  The authors have addressed the main questions/concerns I raised.  Remaining questions are sufficient to be addressed in future work which this paper enables.  The addition of limitations and next steps is also appreciated.

---

### Official Review · Reviewer_Bxrv · 2021-11-02

**Correctness:** 4
**Technical Novelty And Significance:** 4
**Empirical Novelty And Significance:** 4
**Recommendation:** 8
**Confidence:** 3

**Main Review:**

The paper presents a strong and clear theory for the proposed reparameterization. Its advantages over the prior work linear state-space layers (LSSL) are explained very well theoretically and presented empirically.
The convolutional kernel defined by LSSL is a convolutional interpretation to unrolling SSMs over time, granting parallelization to SSMs during training. The proposed model S3 resolves the bottleneck in this formulation by reducing it to a Cauchy kernel, resulting in a significant improvement in the space and computation complexity.
The model is also compared against the state-of-the-art models in a diverse set of benchmarks. Especially the performance in the Long Range Arena (LRA) benchmark is superior.

I have the following questions for the authors.
- The base LSSL model is not included in most of the benchmarks, particularly in the LRA. Is it due to the scalability problem of LSSL?
- Is it expected to get improved results over the LSSL (see Tab. 4 and 5)? Does the proposed reparameterization guarantee a “more” optimal state matrix A?
- The diagonal matrix $\Lambda$ and the vectors $p$ and $q$ are trainable parameters that construct the state matrix A which is initialized to be a HiPPO matrix (Section 3.4). Do these trainable parameters preserve the structure of the matrix A to be HiPPO during and after training? Is it necessary?
- In autoregressive generation tasks (i.e., the model is fed with its own prediction), the sequence models (RNNs, TCNs, etc.) are likely to suffer from error accumulation problem as the prediction horizon increases. Could the authors comment on S3’s performance in a similar task? Can S3 alleviate this problem?


-- Post-rebuttal edit --

I thank the authors for their rebuttal. I read other reviews and the author responses. It is clear that this is both empirically and theoretically a strong paper. The improvements over the baselines are substantial. Looking forward to seeing the follow-up work.

**Summary Of The Paper:**

This paper presents a novel parameterization for the established state-space models (SSM), which tackles the scalability problem in linear state-space layers (LSSL) in modeling long-range dependencies for very long sequences. The proposed technique reparameterizes the structured state matrix in LSSL that allows the state to memorize the past, a key component following the continuous-time memorization theory. A complete theoretical description of this reparameterization and empirical evaluations on a diverse set of benchmarks are presented. The proposed model, S3, achieves astonishing results both in terms of performance and computational efficiency.

**Summary Of The Review:**

This is theoretically and empirically a solid paper. The evaluations show that its superior performance is not limited to modeling long-term dependencies only, but it can be a strong alternative to the established sequence models.

---

> ### Author Response · Authors · 2021-11-20
> **Response to Reviewer Bxrv**
>
> We are encouraged that the reviewer appreciates our theoretical contributions and strong empirical results. We respond to the reviewer’s questions in order:
>
> 1. Most of our results use numbers reported directly from prior work. We included the LSSL where applicable (sequential image classification and speech classification tasks), but the LSSL did not have results on many of our tasks. As the reviewer notes, this is due to the scalability problems of the LSSL; it would not have been able to solve the LRA Path-X task due to memory and computation constraints.
>
> 2. The representations used by the LSSL and S4 are similar, and we do not currently have guarantees on when S4 would perform better. The main advantage of the S4 parameterization is computational.
>
> 3. Our theoretical results show that the previously defined HiPPO matrices are all NPLR. The reviewer asks the natural question about whether the converse holds: are all NPLR matrices also HiPPO operators? We do not currently know the answer to this question, which is a very interesting direction of future study. Empirically, training these parameters does provide a consistent performance increase. (See also response to Reviewer ncZn)
>
> 4. As the reviewer notes, the error accumulation problem is tied to autoregression, and not the sequence model itself. S4 does not address the error accumulation problem per se, but we note that its superior modeling of long-range dependencies may yield more consistent predictions over long horizons. This is an interesting future direction.

---

### Official Review · Reviewer_ncZn · 2021-11-08

**Correctness:** 4
**Technical Novelty And Significance:** 3
**Empirical Novelty And Significance:** 3
**Recommendation:** 8
**Confidence:** 3

**Main Review:**

The paper seems well written both regards to clarity and citations. Contentwise, the theoretical and experimental parts are interesting and relevant. The novel contribution, which is efficient computation of the discretized convolution kernel, is highly technical with details given in the appendix, but the authors did a good job at summarizing the key ideas. On the LRA benchmark, which was originally introduced to benchmark scalable transformer variants, the model sets a new high score on all problems, outperforming transformers on their home turf. The model also outperforms competitors on a raw audio dataset, including a SoTA CNN variant specialized to audio. Overall, the performance is compelling on all considered tasks, in respect of both accuracy and efficiency.

I have just a few questions/remarks:

1) The authors mention in Section 2.2 that linear SSMs can perform poorly due to vanishing/exploding gradients. The given HiPPO matrix is given by the sum of a normal and a low-rank (i.e. NPLR) matrix, and in particular it is not unitary. I wonder then how it gets around the aforementioned problem? I expect the answer can be found by looking up the cited papers, but if there is a short answer to this, it might be good to include it in the discussion.
2) Related to the previous question, it turns out the authors only use the HiPPO matrix as an initialization (although as a very sensible one), but then the algorithm is free to learn any NPLR matrix. Is it true in general for NPLR matrices that they result in well-behaved SSMs in the previous sense, or does that only hold for a neighborhood around the HiPPO matrix? Additionally, I would be interested in what happens (e.g. how performance changes) when a) the HiPPO matrix  $\mathbf{A}$ is fixed throughout the training, and only the parameters $\mathbf{B}, \mathbf{C} \in \mathbb{R}^N$ are trainable, and b) if the model is initialized from a random NPLR matrix, but not the HiPPO?
3) Is the model able to handle multidimensional input signals $\mathbf{u}(t) \in \mathbb{R}^d$?
4) In Section 3.4, it is mentioned that a multidimensional feature map of size H is created by defining H independent copies of S3. I am wondering if it would be more efficient if these copies shared the input-to-state ($\mathbf{B}$) and state-to-state ($\mathbf{A}$) mappings, and only differed in the state-to-feature mapping ($\mathbf{C}$)? Do the authors expect this would negatively affect the results? Somehow it seems wasteful to me for each feature-coordinate $y^i(t)$ to have its own separate $N$-dimensional state representation, instead of sharing a common one (perhaps with a larger state size $N^\prime >> N$).
5) The S3 itself seems to be a linear model, which made me wonder where the nonlinearities are introduced into the deep model? Perhaps, is there an activation placed on the feature map after linearly mixing H independent copies of S3?
6) In Section 4.3 paragraph _Irregularly sampled data_, it seems what the authors really mean is a resolution change. As far as I know, irregular sampling means the data is sampled on a highly non-uniform time grid, which means that in eq.(3) the step size $\Delta_t$ would become time dependent. It looks like this might make the computation of the convolution kernel in eq.(5) a bit problematic (the discretized $\bar{\mathbf{A}}_{\Delta_t}$ matrices might not commute with eachother).

**Summary Of The Paper:**

The paper extends previous work on linear state space models (SSM), where the state transition matrix is fixed to be a highly structured _HiPPO_ matrix, which has provably beneficial properties for memorizing long-term information from continuous-time signals. The main contribution of the paper is regarding the computational aspect of the model. Namely, a novel approach is proposed for computing the convolutional kernel associated to the discretized SSM unrolled over time. This is done by first showing that the state transition matrix can be decomposed as the sum of a normal (i.e. diagonalizable with orthonormal eigenvectors) and a low-rank (rank 1 or 2) matrix, and this representation is then combined with techniques from numerical linear algebra to reduce the problem to computing the Cauchy kernel (i.e. a well-studied problem). The resulting structured state space model (S3) is then placed into a deep neural network setting, and extensive experiments are carried out on various tasks, such as: 1) Long Range Arena, a benchmark collection for scalable transformers, 2) raw speech classification (length-16k audio signals), 3) generative modelling on CIFAR-10 and WikiText-103, 4) sequential image classification on sMNIST, pMNIST and sCIFAR. Overall, the model seems to perform very well on each task either performing close to SoTA or setting a new high score.

**Summary Of The Review:**

This is a solid paper with a novel technical contribution that utilizes nontrivial insights for efficient matrix computations, and with a strong experiments section. The experiments demonstrate not only that the model can be an efficient alternative to transformers on tasks requiring long-range reasoning, but that it also shows promise as a generic sequence model that can be applied across a broad range of tasks. Overall, there are not many drawbacks of the paper for me, other than some unanswered questions in my mind.

---

> ### Author Response · Authors · 2021-11-20
> **Response to Reviewer ncZn**
>
> We are glad the reviewer found the ideas in the paper compelling and clear. The reviewer asked several insightful questions about our proposed model, which we respond to in order:
>
> 1. One answer is that the *discretized* matrix $\overline{A}$ is the one that needs to be close to unitary, and in fact it is by the formula (3). At a higher level, unitary can be considered a side effect and not the main condition needed to handle long dependencies; the HiPPO theory guarantees handling long context by design.
>
> 2. Empirically, we have found that matrices in a neighborhood of HiPPO matrices perform better (see below ablation). This leads to the following interesting question: our theoretical results show that the previously defined HiPPO matrices are all NPLR, but the converse question is an extremely interesting direction for future study.
> (a) The S4 parameterization (Algorithm 1) actually makes it difficult to disentangle learning A from B and C. We note that prior work on the LSSL performed a similar ablation: they found that learning the A matrix provides a consistent boost in performance (e.g., up to 5% on the Speech Commands dataset)
> (b) We performed this ablation using a smaller model on the sequential CIFAR dataset and got 88.38% for HiPPO matrix A, versus 81.36% for random NPLR matrix A. This is still substantially better than non-SSM models, but the HiPPO initialization has a clear advantage.
>
> 3. Multidimensional inputs are handled by the $H$ broadcasting described in Section 3.4 (as well as mentioned in question 4 here). While state spaces technically can handle multidimensional signals (by changing the shape of the $B$ and $C$ matrices), this is actually subsumed by the $H$ broadcasting followed by a linear map to mix the features. Earlier versions of this work did consider higher-dimensional signals, but found no computational or empirical benefit.
>
> 4. The proposed feature sharing was actually considered in earlier versions of this work, but discarded for simplicity. The main reason is that although sharing the A and B parameters does save parameters, it does not save computation. Essentially, the Cauchy kernel needs to be computed for each of the H copies of (A, B, C); even if A and B are constant between each copy, the fact that C is different means each of the H copies still needs to be re-computed separately. Hence there is no computational benefit to sharing some of the parameters.
> As mentioned, this sharing was previously considered; our released code does in fact contain options to share the A, B, or C parameters.
>
> 5. S4 by itself is indeed linear, and nonlinearities are indeed introduced in the depth direction between the S4 and linear layers (Section 3.4). We note that prior work (the LSSL) showed theoretical results explaining why linear RNNs mixed with non-linearities in the depth direction can recover non-linear RNNs.
>
> 6. We agree with the reviewer’s definition of irregularly-sampled data and have changed the title of this subsection in the revised submission.
> We also agree with the reviewer’s point about the computation issue with truely irregularly-sampled data. Indeed, such a setting can still be handled by SSMs, but they will then lose the convolutional representation. One perspective is that in control theory, only linear *time-invariant* (LTI) systems are related to convolutions, while irregularly sampled data would be *time-dependent* as the reviewer points out. Such considerations are interesting directions for future work.

---

> > ### Comment · Reviewer_ncZn · 2021-11-27
> > **Thanks for your detailed answers**
> >
> > Thank you very much to the authors for their detailed answers to my questions. The provided discussion is indeed very interesting and clarify the answer to most of my questions. Although there exists several follow up questions that I would have, this is not a drawback, but a side effect of an intriguing paper. To reiterate, both the theoretical and experimental content are substantial, which clearly warrants publication at the conference. I look forward to seeing further works regarding theoretical analyses and applications of S4.

---

### Public Comment · ~Jason_Rute1 · 2021-11-11
**Can one combine the convolutional and recurrent methods?**

As the reviewers have indicated with their scores, this seems to be a great paper which hopefully lives up to its potential as a fast long range sequence model.

I believe there is a small typo in equation (4).  The last term should have a $\overline{\mathbf{C}}$ in it.

Also, I want to ask if one can combine both the fast recurrent method and the fast convolutional method.  One could imagine for example a document summarization task where one wants to use the convolutional method to process a long document and then the recurrent method to auto-regressively generate a summary of that document.  However, it is not clear if the convolutional calculation gives access to final hidden state $x_N$ which would be needed for this.  Similarly, one might want to perform the convolutional method but with a seed
state $x_0$ which is not $0$.  For example, imagine processing an extremely long sequence in batches with the convolutional method.  One needs to be able to start the next batch where the previous batch left off. Is this possible?

---

> ### Author Response · Authors · 2021-11-20
> **Thanks for the comment**
>
> Thanks for pointing out the typo! It has been fixed in the updated draft.
>
> If we are interpreting your question correctly, you are asking: how can we compute an SSM on a finite "chunk" of the data, using an arbitrary starting state $x_0$, while also computing the final state $x_N$? This calculation is naturally done using the recurrent view, but that loses the parallelization; can the calculation be done using the convolutional view?
>
> The answer to this question is yes, and in fact our released code has this functionality. However, it currently is slower than the vanilla SSM convolution calculation for technical reasons, and additionally may be more prone to numerical instability. We are very interested in the application of this functionality for future work.

---

> > ### Public Comment · ~Mingwei_Ma1 · 2022-06-23
> > **Follow up question**
> >
> > Thanks for the reply.
> >
> > I am another researcher interested in applying your model (which is ingenious!) to our data.
> >
> > Similar to Jason who asked the original question, we are interested in using the CNN version of S4 but due to memory constraints we cannot use the full sequence at inference time, but instead have to divide the sequence into small chunks. For RNN we just keep the final hidden state of each chunk and use that as the starting hidden state of the next chunk and iterate, but we are not sure how to do this for S4.
> >
> > Since you mentioned that your code already has this functionality, could you point out **where it is located and how to use it**? Is it the `step` method for S4 and S4D modules? Also, since you mention "numerical instability", will one obtain the same result at inference time if one uses the chunking method?

---

> > > ### Public Comment · ~Albert_Gu1 · 2022-06-24
> > > **Please use GitHub for code questions**
> > >
> > > The functionality is available in version v1 of the repository: https://github.com/HazyResearch/state-spaces
> > > The outputs of the module will be correct, but it may be more difficult to train a model using TBPTT.
> > >
> > > Please direct all further questions about code to the GitHub issues in the repository.

---

### Author Response · Authors · 2021-11-20
**Response to All Reviewers**

We thank all reviewers for their time and thoughtful feedback. Overall, all reviewers understood the core technical results of our paper and provided perceptive comments and questions about our proposed model. All reviewers appreciated our model’s theoretical and empirical results on long time series across many tasks and data modalities, and provided helpful suggestions for discussion and clarification. Their feedback has helped improve our updated submission, which includes:

- A paragraph on limitations and future directions in the Discussion section
- Improving presentation of the experimental section (e.g. order of tables)
- (Minor) We have additionally renamed our model from S3 to S4

We have responded individually to each reviewer in the direct responses.

---

### Public Comment · ~Albert_Gu1 · 2022-03-09
**Updated revision completed**

To all readers: We have uploaded an updated version of the S4 paper, including the Camera Ready Revision for ICLR 2022. We would also like to note that a version of the paper is on arXiv, which contains additional content that did not fit in the page limits of the ICLR version: https://arxiv.org/abs/2111.00396

This content addresses several recommendations from the reviewers, such as how to interpret the multidimensional and linear aspects of S4. Most importantly, the arXiv version has an additional section that ablates the state matrix A to show the importance of HiPPO and trainability, which several reviewers were curious about. We thank all reviewers and chairs again for their valuable feedback.

---

### Decision · Program_Chairs · 2022-01-20

**Decision:**

Accept (Oral)

**Comment:**

All reviewers agreed this was a very strong submission: it was clearly written, was theoretically and experimentally interesting, and had excellent motivation. A clear accept. Authors: you've already indicated that you've updated the submission to respond to reviewer changes, if you could double check their comments for any recommendation you may have missed on accident that would be great! The paper will make a great contribution to the conference!!